# A refined picture of the native amine dehydrogenase family revealed by extensive biodiversity screening

Eddy Elisée [1], Laurine Ducrot[1], Raphaël Méheust [1], Karine Bastard [2], Aurélie Fossey-Jouenne[1], Gideon Grogan [3], Eric Pelletier [1], Jean-Louis Petit[1], Mark Stam [1], Véronique de Berardinis [1], Anne Zaparucha [1], David Vallenet [1] ✉ & Carine Vergne-Vaxelaire[1] ✉

Native amine dehydrogenases offer sustainable access to chiral amines, so the search for scaffolds capable of converting more diverse carbonyl compounds is required to reach the full potential of this alternative to conventional synthetic reductive aminations. Here we report a multidisciplinary strategy combining bioinformatics, chemoinformatics and biocatalysis to extensively screen billions of sequences in silico and to efficiently find native amine dehydrogenases features using computational approaches. In this way, we achieve a comprehensive overview of the initial native amine dehydrogenase family, extending it from 2,011 to 17,959 sequences, and identify native amine dehydrogenases with non-reported substrate spectra, including hindered carbonyls and ethyl ketones, and accepting methylamine and cyclopropylamine as amine donor. We also present preliminary model-based structural information to inform the design of potential (R)-selective amine dehydrogenases, as native amine dehydrogenases are mostly (S)-selective. This integrated strategy paves the way for expanding the resource of other enzyme families and in highlighting enzymes with original features.

In a time of global need for sustainable manufacturing technologies, the use of enzymes in chemical transformations has become increasingly significant and represents one pillar in the area of white biotechnology[1-9]. Inherent to their mode of action, enzymes are favorable catalysts for the development of environmentally-friendly industrial processes. The search for industrial enzymes that are sufficiently effective and suitable as genuine alternatives to conventional catalysts has become one of the keys to this ecological transition. Application in the green chemical industry depends on the diversity of enzyme activities and on the features added by protein engineering[5]. Hence a diversity of scaffolds is required to broaden the applicability of biocatalysts in industry. Metagenomics provides enzymes from the whole biodiversity, which has still recently been restricted to cultivatable microbial diversity[10,11]. Given access to the DNA of an entire microbiome, it is now possible to explore the biocatalytic potential of the global proteome. Many examples of metagenome-based enzymes have now been described for industrial applications, such as ligninases and xylanases for bioethanol[12]. In reality, many obstacles still have to be overcome to take full advantage of this type of resource for application. In addition to the major issue of successful expression in host organisms for activity screening, in silico selection of high-quality candidates from a huge amount of sequence data is also an important limitation for fast biocatalyst discovery. Such selections are usually made primarily based on sequence identity close to known enzymes and on a restricted amount of data. The variety of templates in terms of sequences and structures that can be obtained by these methods

[1]Génomique Métabolique, Genoscope, Institut François Jacob, CEA, CNRS, Univ Evry, Université Paris-Saclay, 91057 Evry, France. [2]School of Pharmacy, Faculty of Medicine and Health, University of Sydney, Sydney, NSW 2006, Australia. [3]York Structural Biology Laboratory, Department of Chemistry, University of York, Heslington, York YO10 5DD, UK. ✉e-mail: vallenet@genoscope.cns.fr; carine.vergne@genoscope.cns.fr

remains limited, reducing the possibility of identifying enzymes with sufficiently diverse features, such as substrate spectrum or stereopreference. Therefore, efficient bioinformatics strategies for in silico selection are required[13].

One of the enzyme activities for which this diversity is required is the (asymmetric) reductive amination of prochiral ketones with free ammonia. Indeed, the obtained (chiral) amines are found in many active compounds and in the most frequently used chemical intermediates for the production of pharmaceuticals and fine chemicals[14–17]. In addition to the successful application of transaminases in industry[18], enzymes used for amine synthesis from ketones[19–23] include the Amine Dehydrogenases engineered from wild-type amino acid dehydrogenases (eng-AmDHs)[24,25], native AmDHs (nat-AmDHs)[26] recently identified by our group, some reductive aminases (RedAms)[27,28], a subclass of imine reductases (IREDs) active with ammonia, and engineered ε-deaminating L-lysine dehydrogenases[29]. All these enzymes are dependent upon nicotinamide adenine dinucleotide (NAD) cofactors for which different recycling systems are available and effective for biocatalytic applications[30]. To explore the potential of these enzymes, protein engineering, evolution and (meta)genome-based biodiversity screenings have all been reported[31,32]. Despite giving rise to promising biocatalysts, these explorations of biodiversity only considered a limited subset of metagenomic data from which the selection of candidate sequences to be produced was carried out by pairwise sequence alignment to retrieve homologous sequences from reference enzymes. Such approaches enabled the identification of enzymes performing the same targeted reaction with alternative interesting catalytic properties (substrate promiscuity, temperature, pH or solvent stability) and has proved to be quite efficient for this goal. The discovery of the nat-AmDH family by our group using a two-round iteration sequence-based approach, with 2,4-diaminopentanoate dehydrogenases (2,4-DAPDH) as reference enzymes, is another successful example[26]. In this previous work, a family of nat-AmDHs, evolutionarily unrelated to eng-AmDHs, IREDs, and RedAms, has been built from homologous sequences retrieved from the UniProtKB database[33]. These enzymes are (S)-stereoselective with a carbonyl substrate scope largely restricted to short aliphatic aldehydes and methyl-ketones (<6 carbon atoms)[26,34] and are active towards ammonia rather than primary amines. Based on crystallographic structures, this family had been classified through an active site hierarchical tree describing five groups G1-G5. Among the 3D positions P1–P20 defining the active site, position P3 (Glutamate) has been identified as the catalytic residue[26,35]. In a following study in 2020, some other members of this family were added via a limited search in metagenomic sequence databases for marine environments and the human microbiome[34]. Nevertheless, in these two previous studies, not all the diversity present in the publicly available metagenomic databases had been screened due to the lack of an efficient bioinformatic workflow,

reducing the possibility of having an accurate overview of these enzymes among biodiversity and of identifying enzymes with sufficiently diverse features.

To be able to screen a broader representative set of enzyme sequences from biodiversity, we propose here an innovative approach, based on recent developments in bioinformatics. The result is a refined picture of the nat-AmDH family. This method retrieves both close and distant homologs of the already characterized nat-AmDHs by screening several protein databanks including metagenomic ones. In vitro activity data of representative members of the whole nat-AmDH family built into this work are also provided, to attest to the potential of this approach for biocatalytic purposes. In addition, some key enzymes, selected based on in silico considerations, have been highlighted for their activity towards carbonyl substrates not previously reported for this family, thus demonstrating the power of our method to pick-up specific non-usual enzymes within such a huge dataset.

## Results and discussion

The approach used has three main objectives: (i) to cover as widely as possible the sequence space of the nat-AmDH family; (ii) to select enzymes as representative as possible of the biodiversity and (iii) to perform in vitro activity screens to give an overview of the catalytic range of the family (Fig. 1). To meet these goals, we searched for NAD(P)-dependent enzymes and nat-AmDH homologs within billions of sequences using specific HMM profiles. We complemented the nat-AmDH family with distant homologs, namely proteins sharing similar structures and functions with low sequence similarities that are not easily detected using sequence-to-sequence or sequence-to-profile methods. One strategy to collect remote homologs is to screen HMM signatures against other HMM signatures[36]. While the latter strategy has already been used to assign functions to proteins of unknown function[37] or in the large-scale annotation of metagenomic ORFans[38], as far as we know, application for biocatalytic goals has not been reported yet. Both commonplace and more exotic enzymes were selected based on active site analyses, docking experiments and coverage of the AmDH family diversity, before being produced by heterologous expression and tested for their AmDH activity.

### Environmental sampling, clustering and analysis

We collected the data from ten genomic and metagenomic sequence databases, available in 2020, gathering environmental protein sequences from both prokaryotic and eukaryotic kingdoms. Altogether, the biodiversity considered in this work represents nearly 2.6 billion sequence entries. Details about the number of sequences in each database and their respective size are available in Supplementary Table 1. This reservoir of sequences to be screened was restricted to NAD(P)-binding proteins since AmDHs catalyze NAD(P)-dependent reductive amination. To this end, we built from all the considered

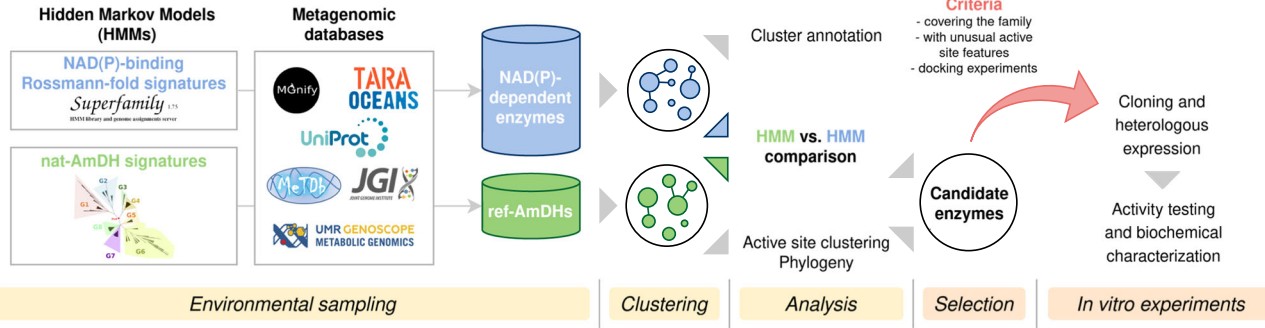

**Fig. 1 | Global strategy for discovering AmDHs among the biodiversity.** This includes five main steps: 1) environmental sampling to define the reference AmDH family (ref-AmDHs) and the NAD(P)-dependent enzyme pool, 2) sequence clustering, 3) cluster analysis including the search for distant homologs by HMM-HMM profile comparison, 4) selection of candidate enzymes and 5) production and in vitro tests of selected enzymes.

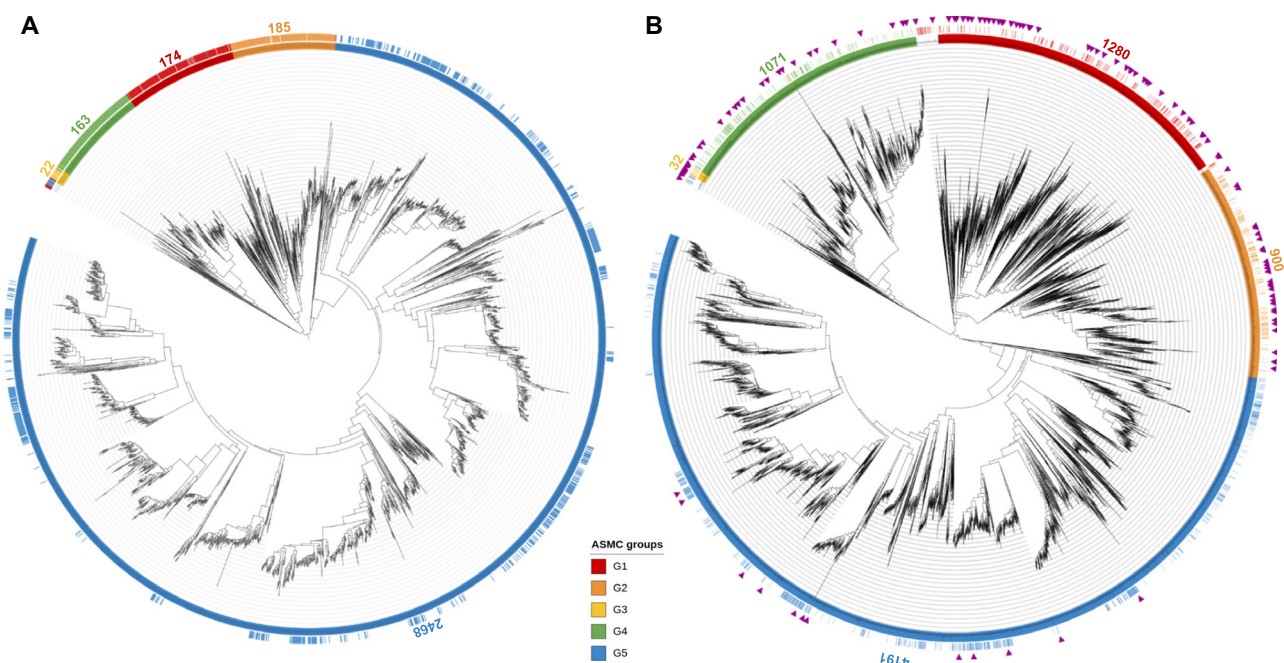

**Fig. 2 | Overview of the ref-AmDHs sequence space.** Phylogenetic trees (removing redundancy at 95% identity on 90% coverage) of (**A**) native AmDHs (nat-AmDHs, from ref. 26, 3032 sequences) and (**B**) extended set of nat-AmDHs (ref-AmDHs, this work, 7,620 sequences) with resulting G1–G5 groups. Colored bars indicate the proteins that were successfully modeled and classified using the ASMC method and purple triangles correspond to the 122 AmDHs tested experimentally in this work. The number of sequences in each group is indicated.

databases a library of 104,734 clusters of homologous proteins (representing 20,097,799 sequence entries) with a Rossmann-fold NAD(P)-binding domain by searching for the SCOP domain NAD(P)-binding Rossmann-fold domain annotation. HMM profiles were then generated for each cluster.

We updated the existing nat-AmDH family (Fig. 2A) by screening the listed (meta)genomic databases with the HMM profile of the nat-AmDH catalytic domain (C-terminus). This search yielded 27,282 AmDH-like sequences that were confirmed to contain an AmDH-like NAD(P)-binding domain, and then reduced to 17,959 sequences by removing redundancy. This set was considered as the extended family of native AmDHs, referred to as "ref-AmDHs" in this work. A phylogenetic tree and an active site clustering (see paragraph « Structural analysis of the ref-AmDH active sites ») were then computed on this extended set of sequences. We built HMM profiles for each phylogenetic (Fig. 2B) and active site (Fig. 3B) group of ref-AmDHs, in addition to one global HMM based on the set of full-length sequences. Figure 2B highlights the extension of the previous G1–G5 groups[26] of the nat-AmDH family (G1: +636%; G2: +386%; G3: +45%; G4: +557%; G5: +70%). A PFAM domain (PF19328, DAP_DH_C) has been created long after the update of the nat-AmDH family (April 2021), modifying the automatic annotation of the AmDH C-terminal domain from dihydrodipicolinate reductase to 2,4-DAPDH (see Supplementary Table 2 and Supplementary Fig. 1). However, the annotation of this domain is still misleading, because not all protein members are expected to catalyze the reduction of 2,4-diaminopentanoate (2,4-DAP) as shown in our previous study[26].

All of the ref-AmDHs HMM profiles were compared afterwards to the NAD(P)H-dependent enzyme families profiles in a HMM-HMM comparison step. On the whole, these different comparisons retrieved the same hits and were then added to the ref-AmDHs set. 440 singleton sequences emerged after a clustering at 80% identity on 80% coverage but no new distinct branches were observed on the phylogenetic tree. However, only 25 of them were considered new, as the remaining 415 had already been found during the AmDH family update but were discarded due to our selection criteria (see Methods). None of these

25 sequences was further considered due to the absence of the Glu residue (position P3) critical for AmDH activity or incomplete sequence of the active site. Hence, although HMM-HMM comparison is a powerful strategy to find distant homologs[36], screening metagenomic databases using a HMM profile was, at least in our case, sufficient to cover the broad diversity of the nat-AmDH family (Supplementary Fig. 2).

Given that remote homology may only be inferred by structure when sequence divergence is high, or that enzyme families with different folds can catalyze the same reaction, we simultaneously attempted to capture additional enzymes using a 3D template-based geometric method called catalophore[39]. Unfortunately, this approach failed to find active site analogs among NAD(P)-binding enzymes, whether the models were derived from PDB or, where appropriate, predicted by the AlphaFold algorithm[40]. Indeed, the few catalophore hits obtained were either the reference AmDHs or false positives in which the match involved a buried region of the protein and not a relevant pocket. This low number of irrelevant results could be explained (i) by the open-form structures obtained using the AlphaFold algorithm[40], in which active sites are distorted, rendering the geometric method ineffective, and (ii) by the absence of a cofactor in the predicted models to direct the 3D search towards the potential catalytic pocket and limit the number of false positives. To remedy this would require significant computational resources (e.g., molecular dynamics, docking) to force the closure state of all divergent models and add the critical nicotinamide cofactor in each of them.

**Structural analysis of the ref-AmDH active sites**
The active sites of the ref-AmDHs were classified to help their comparison, using the Active Site Modeling and Clustering (ASMC) method[41]. It classifies sequences using structural information of protein pockets and predicts functional residues by combining homology modeling, structural alignment and hierarchical conceptual classification. With respect to previous work, the active site pocket is composed of 21 updated key positions named P1-P21 (Fig. 3A, see Methods)[26,35]. Figure 3B presents the hierarchical tree of

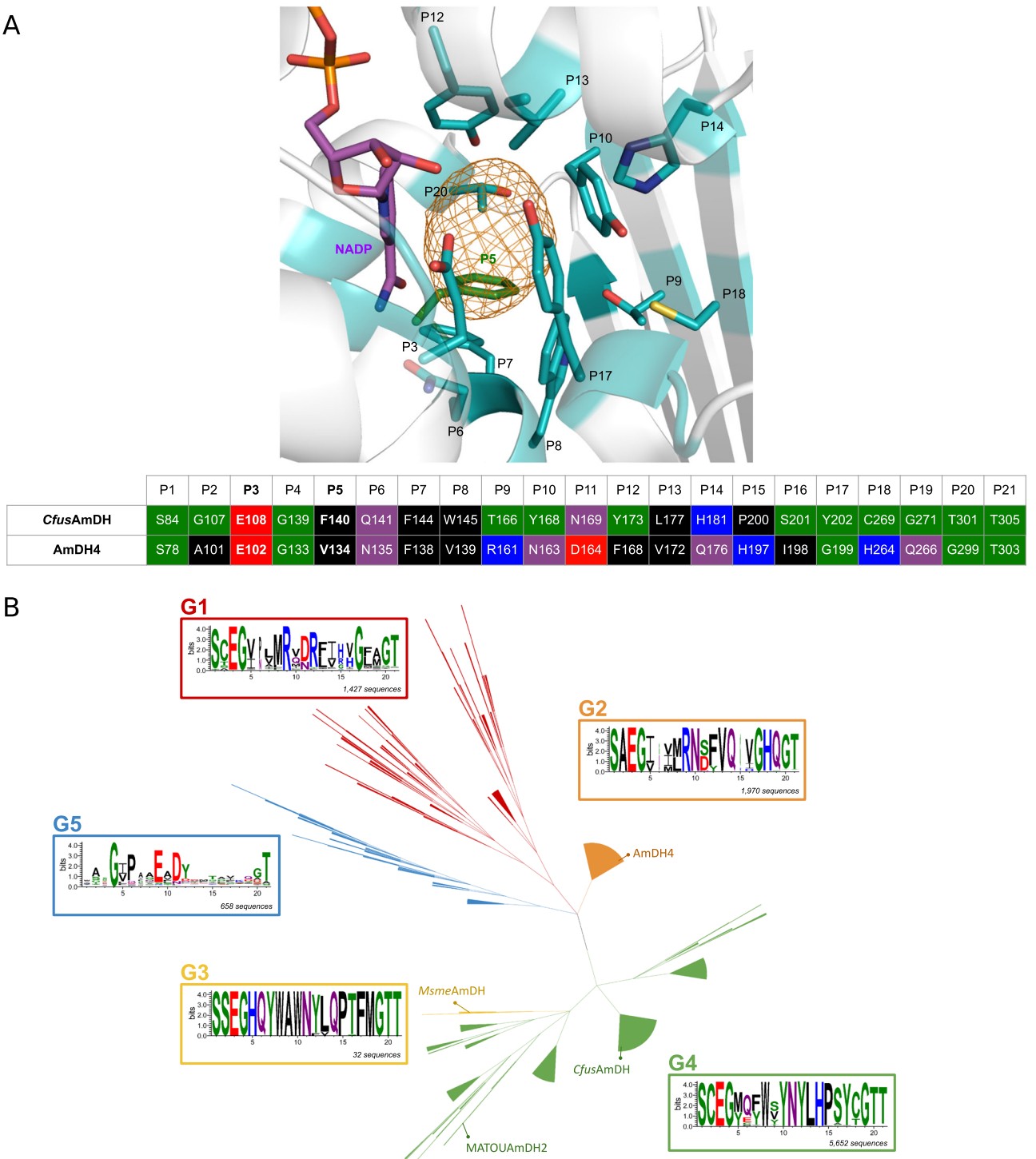

| | P1 | P2 | P3 | P4 | P5 | P6 | P7 | P8 | P9 | P10 | P11 | P12 | P13 | P14 | P15 | P16 | P17 | P18 | P19 | P20 | P21 |
|---|---|---|---|---|---|---|---|---|---|---|---|---|---|---|---|---|---|---|---|---|---|
| *Cfus*AmDH | S84 | G107 | E108 | G139 | F140 | Q141 | F144 | W145 | T166 | Y168 | N169 | Y173 | L177 | H181 | P200 | S201 | Y202 | C269 | G271 | T301 | T305 |
| AmDH4 | S78 | A101 | E102 | G133 | V134 | N135 | F138 | V139 | R161 | N163 | D164 | F168 | V172 | Q176 | H197 | I198 | G199 | H264 | Q266 | G299 | T303 |

**Fig. 3 | Diversity of active sites from the ref-AmDHs family. A** Residues P1–P21 are considered in this study, including the critical catalytic glutamate (P3) and the residue now at position P5 (green), absent from the Mayol et al. analysis. Top: *Cfus*AmDH active site (PDB ID: 6IAU). For greater clarity, only residues closest to the active site pocket (orange mesh) are shown. Bottom: Active site sequences of *Cfus*AmDH compared to AmDH4. For consistency, coloring refers to the WebLogo3 "chemistry" color scheme as described below. **B** Hierarchical tree of the 9763 ref-AmDH active sites, made by the ASMC pipeline. Crystallographic structures used in this work are indicated in their respective ASMC groups. Each sequence logo represents the conservation of the P1–P21 residues. Logos were made using WebLogo3 and its "chemistry" color scheme [green: polar, purple: neutral, blue: basic, red: acidic, black: hydrophobic (charges at physiological pH)].

the ref-AmDHs active site yielded by the ASMC pipeline and on which the five groups (i.e., proteins from G1 to G5 groups) have been mapped, in addition to the existing crystallographic structures. Given the increasing size of G1, G2, and G4 groups (Supplementary Table 3), the active site analysis previously carried out by ref. 26

remains consistent after the addition of metagenomic sequences, revealing more widely conserved consensus residues at each position (Supplementary Fig. 3).

G2 (1970 models) gathers 2,4-DAPDH homologs presumably sharing the same native function in the ornithine fermentation

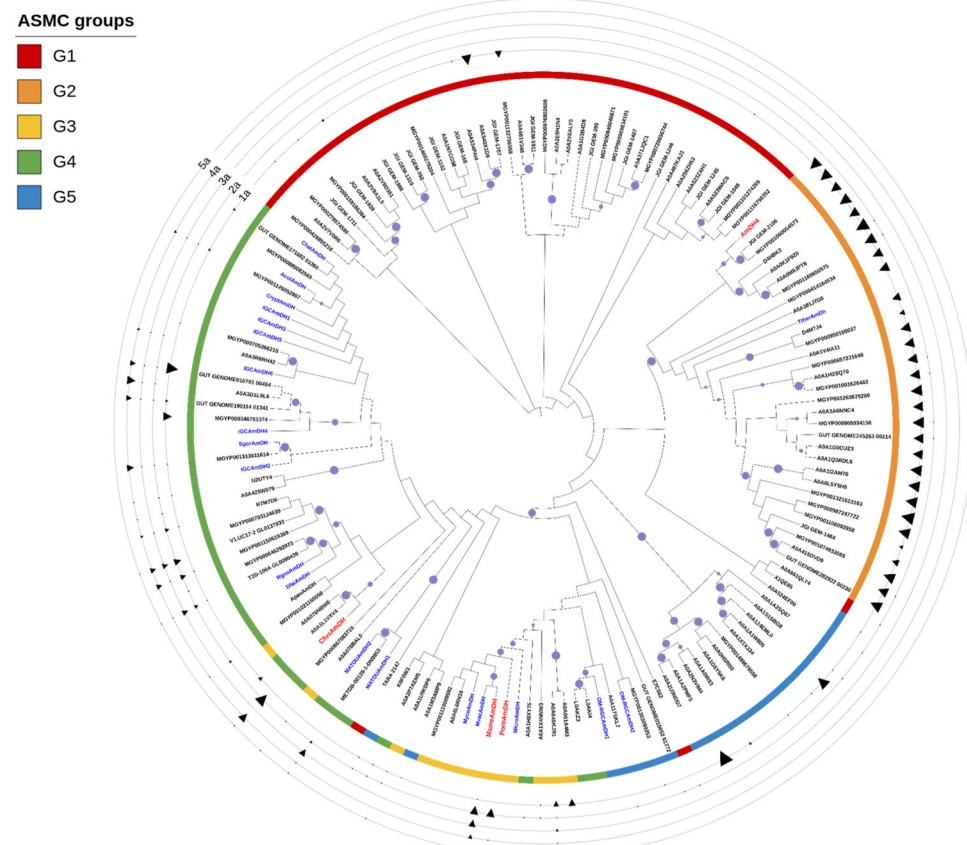

**Fig. 4 | Phylogenetic tree of representatives of the extended nat-AmDH family experimentally tested and their detected activities towards substrates 1a-5a.** Tested reference enzymes (*Cfus*AmDH, *Msme*AmDH, *Porti*AmDH and AmDH4) are indicated in red. Active nat-AmDHs previously reported but not tested in this screening assay are indicated in blue. Bootstrap values > 80% are indicated with purple circles. Analytical yields in **1b-5b** from tested substrates **1a-5a** are shown as black triangles with a size gradient that ranges between 0 and 10 mM.

pathway[42]. The highly conserved P1–P21 and the close sequence similarity of their members support this hypothesis (Supplementary Fig. 3). For G1 (1,427 models), the comparison of its sequence logo with G2 indicated a major difference in position P12 but also a conserved Arg in P9 and His in P15, suggestive of a possible substrate similar to 2,4-DAP with a terminal carboxylic group and an amine reacting group further away in the structure. The conserved Arg in P12 (Phe in G2) suggested coordination with a negatively charged group or a proton acceptor functional group in place of the methyl of 2,4-DAP. Docking experiments with this type of substrates were performed with the model of a G1-enzyme from *Vulcanisaeta distributa* (UniProt ID: E1QRK4) with P1-P21 consensus residues. Computed energies of binding were deemed to be in accordance with a potential reaction ($-5.2$ to $-3.4$ kcal mol$^{-1}$) by comparison with that of 2,4-DAP in AmDH4 ($-5.5$ kcal mol$^{-1}$) (Supplementary Table 4). To complete this P12-based substrate search, a virtual screening, using the list of amines provided in Supplementary Data 1, was performed but did not provide any clues about other potential substrates for this group. Also, given that prokaryotic genes involved in a similar pathway are frequently encoded in a single locus with an operonic organization, an analysis based on conserved genomic context was conducted using NetSyn[43]. However, this did not reveal any clear evidence for the metabolic function of these enzymes (Supplementary Fig. 4).

G3 active sites (32 models) differ more substantially than G4 ones (5652 models), by residue P10, being Trp in G3 and Tyr in G4, by the added residue P5 (His *vs* Met/Tyr) and by residues P14 (Gln *vs* His), P16 (Thr vs Ser) and P17 (Phe *vs* Tyr). In addition to group expansion, differences were substantially highlighted by considering *Cfus*AmDH

as the ASMC reference rather than AmDH4 as reported previously[26]. These groups still share a common branch in the phylogenetic tree (Fig. 2B), highlighting a higher similarity between them compared to other groups.

G5 (658 models) includes enzymes mainly with no catalytic glutamate in P3 and was built considering groups different from G1 to G4. More refined analyses of clusters enabled the selection of specific enzymes (see paragraph « Structure-based selection and activity assays of enzymes with altered substrate scope »).

**In vitro experiments: overview of the biocatalytic activity of the AmDH family**

Given the diversity of the AmDH family, we decided to perform in vitro experiments to demonstrate the reductive amination activity of some representatives. The activity of the 122 selected enzymes over-expressed in *Escherichia coli* was tested at 10 mM substrate loading: (i) common nat-AmDH substrates ((2*R*)-2-amino-4-oxopentanoate (**1a**) (2A4OP) and cyclohexanone (**2a**) in addition, for G3-G5 members, to butan-2-one (**3a**) and furfural (**4a**)) and (ii) some substrates less converted by known nat-AmDHs (hexan-3-one (**5a**), benzaldehyde (**6a**)), to initially identify enzymes with interesting features (Figs. 2B, 4 and 5 and Supplementary Figs. 5 and 6).

For G1 group members, neither **1a**/**1b** and **2a**, nor those hypothesized to fit the active site (Supplementary Table 4), were found to be active (Supplementary Data 2). However, some enzymes were unexpectedly active towards 2,4-DAP (**1b**), the native substrate of G2 members, such as A0A540X1D9, from *Myxococcus llanfairpwllgwyngyllgogerychwyrndrobwllllantysiliogogogochensis*, and

**Fig. 5 | Substrates and products discussed in this work.** The different ketones, aldehydes (**1a-13a**) and corresponding amines (**1b-13b** and **2c-2e**) are represented.

MGYP001132756558. This activity is in accordance with model-based analysis of their active sites (Supplementary Fig. 7). At this stage, the ketone substrates of G1 members are still not known.

For G2 group members, all the tested enzymes were active towards 2A4OP (**1a**), confirming their role in the ornithine degradation pathway (Supplementary Data 3). All the analytical yields in 2,4-DAP (**1b**) were high (7.0–8.6 mM), revealing potential alternatives to AmDH4 used and modified for production of (*S*)−4-aminopentanoic acid from the sustainable levulinic acid[44].

For enzymes of G3 and G4 groups, among the 16 proteins displaying a clear band on SDS gel over the 29 attempts, 13 were active towards at least **2a** (Supplementary Data 4). The latter usually led to the highest amount of amines even if some led to much higher amount of furfurylamine (**4b**) compared to cyclohexylamine (**2b**), including A0A3D1L9L6 from *Clostridiales bacterium* (6.25 *vs* 0.71 mM), GUT_GENOME190114_01341 (4.55 *vs* 0.34 mM) and MGYP001313611614 (4.77 *vs* 0.46 mM). These enzymes appeared to be promising for reductive amination of **4a**, previously described to be a substrate for some RedAms but with primary amines and not ammonia[45]. Interestingly, the enzyme METDB-00128-1-DN9853 displayed activity towards all the tested substrates except **1a**, including **6a** [0.33 mM of benzylamine (**6b**) detected by UHPLC-UV] which was either not or less well converted by other tested enzymes including references. Some enzymes provided high analytical yields of **1b** from **1a**, the substrate of G2 enzymes, especially GUT_GENOME010791_00494 (7.70 mM) and MGYP000346751374 (4.96 mM). Interestingly, these enzymes do not harbor an Arg at P9 as in other G2 members, or an aliphatic residue (Ala/Ile/Val/Ser/Gly/Leu) as in many G3-G4 members (Supplementary Data 5).

As suspected, enzymes from G5 were not active towards the tested substrates, except ones harboring glutamate at P3, such as A0A124EML0 from *Mycobacterium sp. IS-3022*, which was active towards **1a** (Supplementary Data 4). These results confirm a divergent activity for members of the G5 group not harboring the key glutamate in P3.

The sequence identity matrix of the 72 active representative AmDHs revealed substantial diversity within this updated set of experimentally validated nat-AmDHs, which included the previously characterized ones (Supplementary Data 6). Particularly, for G3, active enzymes displayed only 34–49% sequence identity with those previously characterized from the same group (*Micro*AmDH/*Msme*AmDH/*Porti*AmDH) and <36% and 29% with *Cfus*AmDH and MATOUAmDH2, respectively. For G4, all the active proteins displayed less than 55% sequence identity with the G4 reference *Cfus*AmDH, and less than 41% and 33% with *Micro*AmDH/*Msme*AmDH/*Porti*AmDH and MATOUAmDH2, respectively. Such wide sequence homology could not have been obtained by protein engineering, thus emphasizing the benefit of this type of workflow.

## Structure-based selection and activity assays of enzymes with altered substrate scope

In addition to providing an overview of the nat-AmDH biocatalytic activity, we decided to use the large diversity obtained through this work to search for AmDHs with specific P1-P21 residues that might alter the substrate scope (Supplementary Figs. 8, 9 and 10 and Supplementary Data 7).

Based on previous results, mutations into alanine at position P5 and P8 facilitate the accommodation of more sterically demanding substrates (6–10 carbon atoms)[27]. Thus, we selected and heterologously produced 17 ref-AmDH enzymes harboring small residues at positions P5 or P8 (Ala/Val/Gly/Leu/Ile/His/Ser/Thr) or for which the models displayed an apparent larger active site pocket. Except for A0A138ZYM0, from *Gonapodya prolifera*, all the members satisfying this criterion come from metagenomic databases and mainly eukaryotic ones, once again underlining the value of the workflow used. All of them were confirmed to have AmDH activity with analytical yields between 8.5 and 69.8% in the reference product **2b**, except for MGYP001470669209 for which a His at P5 and/or a Gln in place of His at P14 may be detrimental for activity. Activity towards bulkier substrates hexanal (**7a**), octanal (**8a**) and 4-phenylbutan-2-one (**9a**) was

**Table 1 | Analytical yields and enantiomeric excess in bulky amines 8b-11b**

| | heptanamine (10b) | octanamine (8b) | heptan-2-amine (11b) | | 4-phenylbutan-2-amine (9b) | |
|---|---|---|---|---|---|---|
| | conv. (%) | conv. (%) | conv. (%) | ee (S) (%) | conv. (%) | ee (S) (%) |
| METDB_03 | 33.4 ± 0.9 | 18.5 ± 0.7 | 16.0 ± 0.7 | 96.5 ± 0.1 | 67.9 ± 0.0 | >99.8 |
| METDB_02 | 26.3 ± 1.6 | 14.1 ± 0.1 | 19.4 ± 0.9 | 98.2 ± 0.3 | 63.3 ± 4.5 | >99.8 |
| *Cfus*AmDH-W145A | 48.3 ± 0.5 | 28.8 ± 0.5 | 44.5 ± 1.6 | >99.8 | 78.7 ± 0.7 | >99.8 |
| *Cfus*AmDH | 6.3 ± 0.2 | nd | 0.1 ± 0.0 | | nd | |
| without enzyme | nd | nd | nd | | nd | |

*nd* not detected; empty cell: not tested. Reactions conditions: 10 mM substrate, 2 M NH$_4$HCO$_2$ buffer, pH 9.0, 0.2 mM NADP⁺, 0.2 mM NAD⁺, 11 mM glucose, 3 U ml⁻¹ GDH-105, 1.0 mg ml⁻¹ purified enzyme, 24 h, 30 °C. Analytical yields in amines and ee were obtained after derivatization with BzCl and FDAA respectively, and UHPLC-UV analysis (conditions 1) (see Methods). Uncertainties represent the range of values obtained with two independent experiments. Chromatograms are given in Supplementary Figs. 11 and 12. Source Data are provided as a Source Data file.

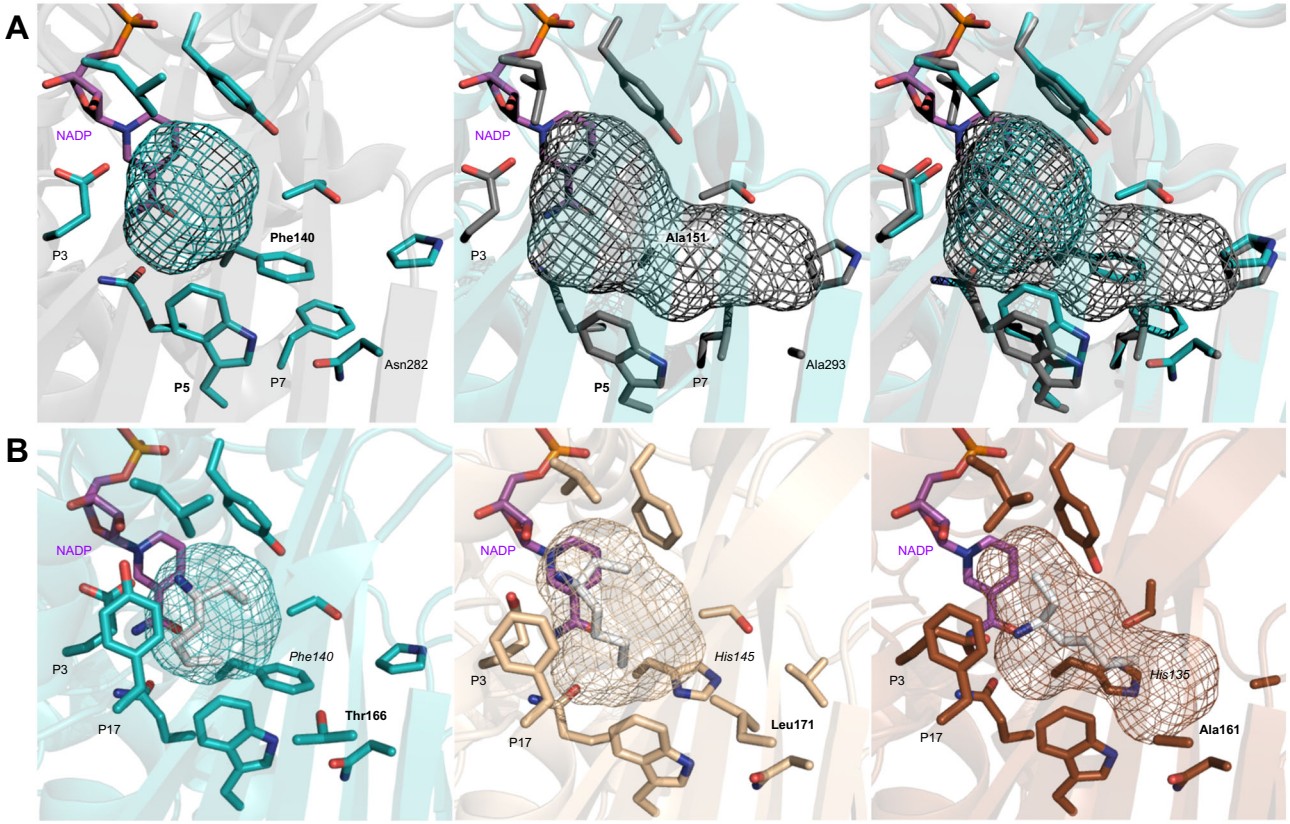

**Fig. 6 | PyMOL visualization of some nat-AmDHs active site. A** Active site cavities of *Cfus*AmDH (left, PDB ID: 6IAU, chain **B** and METDB-03 (middle, *Cfus*AmDH-based homology model). Their superimposition (right, RMSD = 0.21 Å) highlights the larger cavity of METDB-03, due to the F140/A151 replacement and the W145/W156 displacement, compared to *Cfus*AmDH; **B)** Positions P5 (italic) and P9 (bold) responsible for pocket enlargement of MGYP000211951848 (right—*His135*, **Ala161**) relative to *Cfus*AmDH (left—*Phe140*, **Thr166**) and A0A229HGK2 (middle—*His145*, **Leu171**). For the sake of clarity, only one correct conformation of docked (3*S*)-heptan-3-amine (white) for each enzyme is shown.

detected for METDB-02 and METDB-03 and was then confirmed with purified enzymes with analytical yields up to 67.9% in **9b** for METDB-03, the same order of magnitude observed with those obtained with the P8-mutant *Cfus*AmDH-W145A (Table 1, Supplementary Table 5 and Supplementary Figs. 11 and 12)[35]. These two native enzymes, coming from *Litonotus pictus*, are complementary to (*R*)-selective eng-AmDHs and RedAms such as TM_PheDH[46] and *At*RedAm[47], as they generate the opposite enantiomer of the product (*S*)−4-phenylbutan-2-amine ((**S**)−**9b**). This (*S*)-stereoselectivity, already observed for members of this family, is a characteristic maintained in this extended group differentiating these enzymes from the other NAD(P)-dependent enzymes performing reductive amination. Moderate activities were also measured with heptanal (**10a**) and the aliphatic ketone heptan-2-one (**11a**). From a structural viewpoint, the active site of METDB-03 should be wider and could accommodate bulkier substrates than those accepted by *Cfus*AmDH (150 Å vs 63 Å). Ala151 (P5) opens the cavity, together with Trp156 (P8) being slightly moved away due to Leu155 (P7) which occupies the space left by Ala293 (Asn282 in *Cfus*AmDH) in the second sphere (Fig. 6A). Compared to METDB-03, this space gain is limited in the METDB-02 active site (119 Å) by the presence of Val151 (P5) and Ile177 (P9), instead of Ala151 and Val177, respectively, which is in accordance with the in vitro results with slightly lower analytical yields in **8a-11a**. We confirm here that P5 is a critical residue to accommodate more sterically demanding substrates. In our previous study, this hypothesis could not be studied further due to the instability of some P5-mutants[35]. Once again, these conclusions suggest that the wealth of different characteristics obtained from natural diversity is substantial and worth considering.

**Table 2 | Analytical yields and enantiomeric excess in 3C-amines 5b, 12b-13b**

| | hexan-3-amine (5b) | | heptan-3-amine (13b) | | hexan-2-amine (12b) | |
|---|---|---|---|---|---|---|
| | conv. (%) | ee (S) (%) | conv. (%) | ee (S) (%) | conv. (%) | ee (S) (%) |
| MGYP000211951848 | 45.5 ± 4.8 | 53.3 ± 0.4 | 23.7 ± 2.7 | − 29.3 ± 3.7 | 88.3 ± 0.4 | 96.3 ± 0.2 |
| MGYP001209562846 | 50.1 ± 2.3 | 68.6 ± 2.3 | 41.5 ± 2.5 | −25.4 ± 3.4 | 85.9 ± 0.0 | 95.2 ± 0.2 |
| A0A229HGK2 | 63.3 ± 2.7 | >99.9 | 96.5 ± 8.0 | 99.9 ± 0.1 | 99.1 ± 0.0 | 99.4 ± 0.1 |
| A0A1Q4UXH9 | 61.2 ± 0.1 | >99.9 | 30.7 ± 0.7 | 93.4 ± 0.1 | 98.6 ± 0.3 | 99.0 ± 0.1 |
| A0A365ZD63 | 35.2 ± 1.8 | 79.8 ± 0.1 | 6.5 ± 0.1 | 37.9 ± 1.3 | 96.4 ± 0.6 | 98.9 ± 0.1 |
| A0A646KJR1 | 43.8 ± 2.2 | 99.0 ± 0.2 | 44.2 ± 0.3 | 88.3 ± 0.5 | 96.3 ± 0.2 | 96.4 ± 0.1 |
| *Micro*AmDH | 65.3 ± 0.6 | 96.2 ± 0.1 | 13.4 ± 1.2 | 96.9 ± 0.2 | 94.7 ± 0.0 | 87.6 ± 0.1 |
| *Porti*AmDH | 25.2 ± 0.6 | 64.8 ± 1.4 | 9.3 ± 0.6 | − 25.5 ± 1.5 | 88.9 ± 1.4 | 96.6 ± 0.1 |
| *Cfus*AmDH | 7.8 ± 0.4 | 38.0 ± 1.3 | 0.1 ± 0.1 | − 38.2 ± 0.3 | 86.2 ± 0.0 | 97.7 ± 0.1 |

nd not detected. Reactions conditions: 10 mM substrate, 2 M $NH_4HCO_2$ buffer, pH 9.0, 0.2 mM $NADP^+$, 0.2 mM $NAD^+$, 11 mM glucose, 3 U ml⁻¹ GDH-105, 1.0 mg ml⁻¹ purified enzyme, 24 h, 30 °C. Analytical yields in amines and ee were obtained after derivatization with BzCl and FDAA respectively, and UHPLC-UV analysis (conditions 1) (see Methods). Uncertainties represent the range of values obtained with two independent experiments. Chromatogram and calibration curves are given in Supplementary Figs. 14 and 15. Source data are provided as a Source Data file.

The ketones bearing the carbonyl function at C3 of the carbon chain of acyclic compounds (3C-ketones) was another class of target substrates, this ketone position being accepted by a minority of previously studied nat-AmDHs (*Msme*AmDH, *Micro*AmDH, *Porti*AmDH)[48]. We selected and overproduced 29 selected enzymes based on the structural hypotheses detailed in Supplementary Fig. 13, using the positive activity of A0A646KJR1 from *Streptomyces jumonjinensis* towards hexan-3-one (**5a**) detected in this study as a basis (Supplementary Data 4). Except for A0A4S3B2N2 (*Vagococcus silagei*), all the 14 enzymes displaying analytical yields above 2% with the reference substrate **2a** displayed activity towards **5a** (Supplementary Table 6). This activity, which was highest for MGYP000211951848, MGYP001209562846, A0A229HGK2 (*Streptomyces sp. NBS 14/10*), A0A1Q4UXH9 (*Streptomyces uncialis*), A0A365ZD63 (*Prauserella sp. PE36*) and A0A646KJR1, in addition to *Micro*AmDH and *Porti*AmDH to a lower extent, was confirmed on purified enzymes with analytical yields up to 65.3 % for hexan-3-amine (**5b**) and 96.5 % for heptan-3-amine (**13b**) with A0A229HGK2 (Table 2, Supplementary Figs. 14 and 15). Again, (*S*)-stereoselectivity predominates, but is not exclusively observed for some enzymes. Docking experiments gave results in accordance with the conversion rates in **13b** and the unusual observed (*R*)-selectivity observed with MGYP000211951848, MGYP001209562846 and *Porti*AmDH. Structural models suggested that MGYP000211951848 harbors a larger pocket than A0A229HGK2 and *Cfus*AmDH, mainly due to the smaller residue Ala161 (P9) replacing Leu171 and Thr166, respectively, thus enabling the long carbon side chain of the amine to be accommodated (Fig. 6B, Supplementary Data 8). Among some other amination enzymes reported for 3C-ketones, Ch1-AmDH was described to catalyze the formation of the opposite enantiomer (*R*)-hexan-3-amine and Rs-PhAmDH or *Gk*AmDH-M3/M8 gave (*R*)-1-phenylalkan-3-amine derivatives from the relevant ketone substrates[49,50]. *Nf*RedAm and *Nfis*RedAm afforded 90% and 52% of (3*R*)-octan-3-amine but with only 40% and 58% *ee*, respectively[28]. The discovered AmDHs clearly complement the previous low number and diversity of NAD(P)-dependent enzymes active towards 3C-ketones. Activity towards hydroxyl-functionalized methyl ketones, studied by refs. 51,52 with engineered AmDHs, could be presumed based on the activity of *Msme*AmDH and *Micro*AmDH towards hexan-3-one in addition to 1-hydroxy-propan-2-one and 1-hydroxy-butan-2-one[53].

The availability of nat-AmDHs capable of converting substrates bulkier than $NH_3$ would open up biocatalytic possibilities to access substituted amines with this family of enzymes. The model of *Cfus*AmDH docked with cyclohexanone and ammonia clearly identified the P13 residue (L177 in *Cfus*AmDH equivalent to L180 in MATOUAmDH2) as the first sphere of the active site ceiling that could limit the size of the amine substrate (Supplementary Fig. 16)[54]. Within the ref-AmDH enlarged set, 16 of the 17 selected enzymes bearing a smaller residue than Leu at P13 (Val/Thr/Ile/Ala) displayed AmDH activity (activity against **2a** with ammonia (**b**)) (Supplementary Table 7). The 10 hits active with methylamine (**c**) were confirmed with purified enzymes giving high analytical yields (65.8−89.0%) for *N*-methylcyclohexylamine (**2c**), thus surpassing the results of previously described nat-AmDHs and of the mutant MATOUAmDH2-L180A (Fig. 7, Supplementary Fig. 17). None of them gave satisfactory analytical yields with ethylamine (**d**) but notable activities were measured with the more constrained cyclopropylamine (**e**), particularly with A0A365ZD63, which gave 66.0% analytical yield of *N*-cyclopropylcyclohexylamine (**2e**). Interestingly, analytical yields were still high with only 2 equivalents of methylamine (**c**) donor, corroborating the proposed catalysis of both imine formation and imine reduction by nat-AmDHs. These enzymes could be complementary to *Nf*RedAm, *Ad*RedAm, Ch1-AmDH and Rs-AmDH, which mainly form (*R*)-methyl/ethylamines, even if their activities towards aromatic and acyclic aliphatic ketones, reported to be transformed by the latter, remain to be studied[55,56]. Structurally, 9 of these 10 enzymes harbor a threonine residue at P13. Docking experiments of the *N*-methylcyclohexyliminium intermediate provided higher energies of binding for the non-active enzyme IGC-32 (−5.80 kJ mol⁻¹) compared to active ones (−7.51 to −7.11 kJ mol⁻¹), with the intermediate in a flipped position. The isoleucine residue (Ile175, P13) is too bulky, thus preventing the good positioning of the amine/iminium with P3 and the C4 atom of the nicotinamide ring of NADP (Supplementary Figs. 18 and 19 and Supplementary Table 8). On the whole, A0A365ZD63 turned out to be a key enzyme both for expanding amine substrate scope and also for the transformation of 3C-ketones.

Focusing on these substrate scope specificities, we have highlighted in this work 17 AmDHs that constitute very promising biocatalysts and/or templates for further studies. Interestingly, their similarity in terms of sequence shows correlation with their biocatalytic features, even though the selection criteria were only based on structural characteristics (Supplementary Data 9).

## Cofactor specificity of nat-AmDHs

Having in hand some experimental data for the cofactor preference of certain nat-AmDHs, we compared the key residues interacting with the adenosine ring, the hydroxyl at the position 2' of the ribose ring (2'OH), in NAD, or its phosphorylated form (2'P), in NADP. At first sight, one observed that many NADH-nat-AmDHs harbor a glutamate at position 36 (*Cfus*AmDH numbering) instead of a smaller residue (Ala or Ser) for NADP-dependent nat-AmDHs. This is in accordance with the already

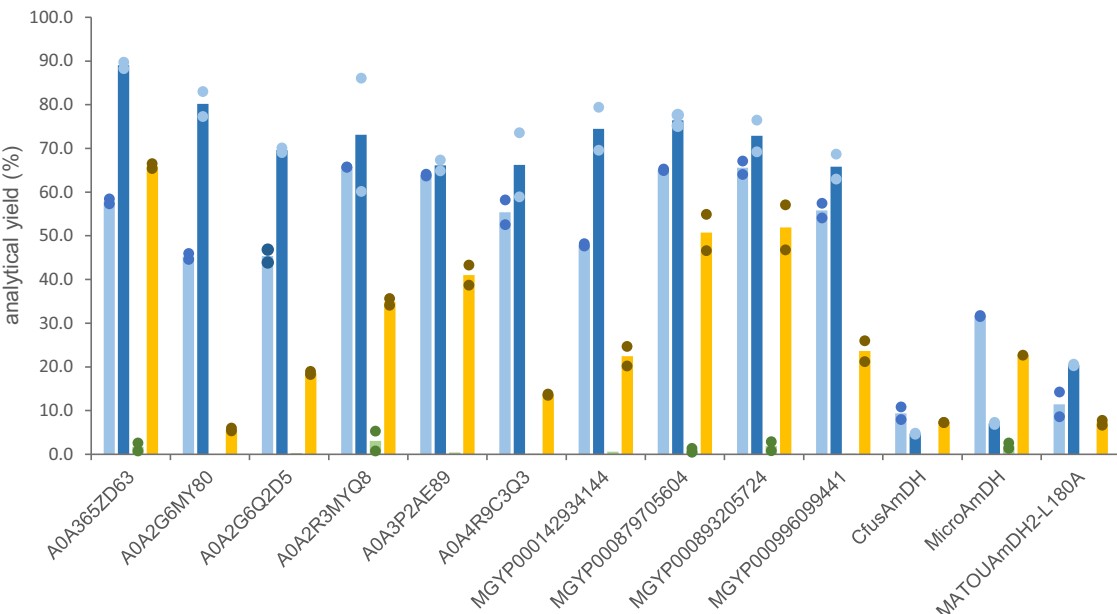

**Fig. 7 | Analytical yields in *N*-alkylamines 2c-2e.** Reactions conditions: 10 mM substrate, 200 mM TRIS.HCl buffer pH 9.0, 250 mM amine donor **c-e** (or 20 mM), 0.2 mM NADP + , 0.2 mM NAD + , 11 mM glucose, 3 U ml⁻¹ GDH-105, 1.0 mg ml⁻¹ purified enzyme, 24 h, 30 °C. Amounts of amines **2c-2e** were obtained after derivatization with BzCl and UHPLC-UV analysis (conditions 2) (see Methods). Bars represent the average of values obtained with two independent experiments (*n* = 2; dot plots) for the reaction of **2c** with 20 mM of **c** (light blue), **2c** with 250 mM of **c** (dark blue), **2d** with 250 mM of **d** (green) and **2e** with 250 mM of **e** (yellow). Chromatogram and calibration curves are given in Supplementary Fig. 17. Source data are provided as a Source Data file.

reported consensus that NADH-enzymes display a negatively charged residue at C-ter of the β2 strand compared to NADP-dependent enzymes in which such a negative charge may disrupt the correct binding of the phosphorylated ribose[57].

Taking advantage of the data gathered on the whole nat-AmDH family, we decided to further detail the occurrence of residues close to the key position 36 by focusing on residues 36 to 41 (*Cfus*AmDH numbering), hereafter named R1 to R6. We performed a multiple sequence alignment of a non-redundant subset of ref-AmDHs followed by a sequence-based clustering, under key position R1, generating 13 clusters (Supplementary Fig. 20). Analysis of these clusters supported by structural modeling, experimentally validated results[34,58] and reported hypothesis[57] (see Supplementary Discussion), led to the following conclusions. NADH-nat-AmDHs (e.g., AmDH4) should bear a negatively-charged residue (Asp, Glu) in the R1 position and an aliphatic (Val, Ile) or bulkier (Arg, Tyr, Phe) one in R2, R3 and R6 positions to block interaction with or placement of the 2'P group of the NADP cofactor. Secondly, NADP-dependent nat-AmDHs (e.g., *Msme*AmDH) should have a short-chain residue (Ala, Ser) in R1 position, a positive charge (His, Arg, Lys) in R2 and R6 positions, and a polar (Ser, Asn) residue in R3 position. Finally, nat-AmDHs able to accept both NAD and NADP (i.e., *Cfus*AmDH) should display a mix of NAD- and NADP-dependent enzyme features, namely to have a negative charge (Asp, Glu) in the R1 position, a positive charge (His, Arg, Lys) in R2 and R6 positions, and a polar (Ser, Asn) residue in the R3 position.

This bioinformatic analysis on a large set of enzymes can be an alternative to generation of libraries of variants proposed by the online tool "Cofactor Specificity Reversal—Structural Analysis and Library Design" (CSR-SALAD)[59], which was implemented by Nestl and coworkers to alter the nicotinamide cofactor specificity of the (*R*)-selective IRED from *Myxococcus stipitatus*, focusing on the positions Asn32, Arg33, Thr34, Lys37[60].

In general, this work enabled a considerable advance in the knowledge of AmDH activity within biodiversity, providing a comprehensive picture of the nat-AmDH family that greatly extends the diversity of the biocatalyst portfolio for amine synthesis. Indeed, the substrate spectrum of some nat-AmDHs described in this work and their very low homology with the previously reported nat-AmDHs opens the door to numerous applications in synthesis and provides avenues for structural studies. In addition, the recent rise of computational approaches to accurately predict protein structures could help to expedite structural studies of enzyme families with few experimentally determined 3D structures[40,61,62].

In the end, the bioinformatic workflow set up in this work and supported by in vitro experiments is a powerful strategy for widely screening biodiversity and drastically increasing the number and diversity of biocatalysts. This would not have been achievable by restricting ourselves to genomic databases or a limited number of metagenomic sampling. This workflow can be used directly for other NADP-dependent oxidoreductases benefiting from the NADP-dependent enzymes already collected from (meta)genomics databases, and we are currently in the process of generalizing it to allow its applicability to other families of enzymes. For biocatalytic goals, this diversity, mainly brought by metagenomic databases, can be used to find unusual sequences or active sites leading to particular features, as exemplified in this study.

## Methods
### Genomic and metagenomic databases
Protein sequences were retrieved from different databases: UniProtKB[33] (SwissProt; TrEMBL), GEM[63] (Genomes from Earth's Microbiomes), UHGP[64] (Unified Human Gastrointestinal Protein), MGnify[65] (EMBL-EBI), IGC[66] (Integrated Gene Catalog of Human gut), MetDB[67] (Marine Eukaryotes Transcriptomes), OM-RGC[68] (Ocean Microbial Reference Gene Catalog), SMAGs[69] (Tara Oceans Eukaryote Metagenome Assembled Genomes) and MATOUv2[70] (Tara Oceans Eukaryote Gene Catalog). Those were downloaded using either a File Transfer Protocol (SwissProt, TrEMBL, UHGP and MGnify), or an online portal (GEM, IGC, OM-RGC, MetDB). SMAGs and MATOUv2 are in-house databases built from Tara Oceans' expeditions. Further details, such as web links, are available in Supplementary Table 1.

## Rossmann-fold NAD(P)-binding domain signatures in metagenomics databases

Amine dehydrogenases display a N-terminal Rossmann-fold NAD(P)-binding domain that we searched for in metagenomics databases using the hmmsearch tool (HMMER[71] package, version 3.3) and the SCOP Superfamily signature (SSF51735, 301 different HMMs). All sequences with at least one match (score ≥50, see details in Supplementary Methods) with one of these HMMs were further considered in a second filtering step by running the Superfamily assignment script (superfamily.pl, details here https://supfam.mrc-lmb.cam.ac.uk/SUPERFAMILY/howto_use_models.html). Then, every sequence with a SSF51735 annotation was kept as it is the best annotation for the corresponding domain, among all available SCOP domain signatures[72]. The distribution of collected sequences is reported in Supplementary Table 9.

## NAD(P)-binding enzyme clustering

NAD(P)-binding enzyme clustering was carried out using a two-step procedure described in ref. 73. and resumed hereafter. Protein sequences were clustered into families using the greedy set cover algorithm from MMseqs2 software[74] (version 12.git113e321, parameters: -s 7.5 -e 0.001 -c 0.8 --cov-mode 0 --min-size 2). Secondly, proteins of each family were aligned, using the result2msa parameter of MMseqs2, and HMM profiles were generated from those multiple sequence alignments, using the HHpred suite[75] (v3.0.3). Those HMM profiles were involved in the search for nat-AmDH distant homologs. Families were then compared to each other using HHblits[76] (v3.0.3, parameters: -v 0 -p 50 -z 4 -Z 32000 -B 0 -b 0). A similarity score (probability × coverage) was applied to weight the input network in the final clustering done by the Markov Clustering algorithm[77] (parameters: --probs 0.95 --coverage 0.75 -I 2). The resulting 1098 NAD(P)-binding enzyme superfamilies were then annotated using well-known domain and sequence signatures (Pfam, KEGG, TMHMM, SignalP).

## Nat-AmDHs signature in metagenomic databases

The number of AmDH sequences in the previously published ASMC set (G1-G5 groups; 2,011 sequences)[26] was reduced to 1816 sequences by removing obsolete sequences (134 sequences reported as such on the UniProt website) as well as those containing less than 250 or more than 500 amino acids while checking for the presence of P1-P20 positions (61 sequences removed). These 1816 sequences were then aligned using the MAFFT[78] sequence alignment software (v7.310, auto mode). The AmDH4 sequence (UniProt ID: A9BHL2) was used as a reference to split the resulting multiple sequence alignment in two parts and obtain two HMM profiles, one for the N-terminal Rossmann-fold NAD(P)-binding domain and one for the C-terminus catalytic domain. Given the InterPro annotation of its NAD(P)-binding domain (IPR036291 entry, residues 1-145), we considered Ile145 as the cutoff residue after which the catalytic domain starts. The two multiple sequence alignments were transformed into HMM profiles with the hmmbuild tool (HMMER[71] package, v3.3). Screenings were performed using the hmmsearch tool (HMMER[71] package, v3.3). The threshold selection procedure and the distribution of collected sequences are detailed in Supplementary Methods and Supplementary Table 10. Redundancy (27,282 to 17,959 AmDH-like sequences) was removed using the CD-HIT[79,80] (v4.6) clustering algorithm at 100% of sequence identity to obtain the set of 17,959 ref-AmDHs.

## Active site analysis and phylogeny of the ref-AmDHs

The non-redundant set of ref-AmDHs was submitted to ASMC software[41] as described hereafter. A homology modeling step was performed using four available template structures: AmDH4 (PDB ID: 6G1M, chain B)[26], CfusAmDH (PDB ID: 6IAU, chain B)[26], MsmeAmDH (PDB ID: 6IAQ, chain A)[26] and MATOUAmDH2 (PDB ID: 7ZBO)[54]. Proteins sharing at least 23% of sequence identity with AmDH4,

CfusAmDH, MsmeAmDH or MATOUAmDH2, i.e., 9,886 proteins, were modeled. Their active sites were defined by the updated 21 CfusAmDH residues named P1-P21 resulting from the addition of a position between P4 and P5 positions[35]. In AmDH4-like ones (G2 group), residues Pro136 and Leu140, formerly P5 and P6, were replaced by residues Phe138 and Val139 as P6 and P7, respectively. All 9,886 models were superimposed on the CfusAmDH structure to extract all the residues aligned with the 21 residues of the reference pocket and build a structure-based multiple sequence alignment of as many sequences. Finally, a sorting step, using WEKA algorithm, was carried out to classify and generate a hierarchical tree of 9763 active sites in which only 15-member clusters were retained. Alternatively, the same non-redundant set was reduced. This was used to construct a sequence-based multiple sequence alignment in which misaligned regions were removed before designing a phylogenetic tree using MAFFT[78] (v7.464), TrimAl[81] (v1.2) and IQ-TREE[82] (v1.6.12) softwares, respectively. Phylogenetic trees were visualized and printed using the Interactive Tree of Life (iTOL) online tool[83].

## Distant homology through HMM-HMM comparison

In order to search for nat-AmDH distant homologs, we compared HMM profiles from nat-AmDH family to those from NAD(P)-binding enzyme families using HHblits[76] (parameters: -v 0 -p 50 -z 4 -Z 32000 -B 0 -b 0). Families were considered as hits if probability scores were greater than or equal to 95%.

## Selection of representative nat-AmDHs to be screened

Within the set of ref-AmDHs, 122 members that cover each group of the ref-AmDHs ASMC (7,039 sequences) were selected based on three main criteria: 1) presence of the catalytic glutamate in P3; 2) phylogenetic tree coverage; 3) predicted solubility of the proteins[84]. Supplementary Data 5 details their sequence ID and P1-P21 positions. The enzymes were overexpressed in Escherichia coli and tested as crude cell-free extracts (Supplementary Figs. 5 and 6) as described below.

## In vitro experiments: general

All the chemicals were purchased from commercial sources and used without additional purification. UHPLC analyses were performed on a UHPLC U3000 RS 1034 bar system (Thermo Fisher Scientific) equipped with a UV detector using a Kinetex® F5 (Phenomenex) column (100 × 2.1 mm; 1.7 µm). Spectrophotometric assays were recorded on Spectramax® Plus384 Molecular Devices with 96-microwell plates.

## Production of enzymes

The selected genes were synthesized by Twist Bioscience (San Francisco, United States) and optimized for expression in Escherichia coli. Genes were then amplified from these synthetic fragments by adding to the primers (Supplementary Data 10) specific extensions for cloning into pET22b(+) (Novagen) modified for ligation-independent cloning (LIC). The forward primers introduced a hexahistidine tag sequence in the proteins after the initial methionine for purification purposes. The cloned genes were sequenced. The verified constructions were then transformed into E. coli BL21-CodonPlus (DE3)-RIPL competent cells (Agilent Technologies) for induction. These were grown on Terrific Broth (TB) medium containing 0.5 M sorbitol, 5 mM betaine and 100 µg mL⁻¹ carbenicillin at 37 °C until reaching an OD600 of 0.8–1.2 (1.8–2.0 for batches purified by tandem with gel filtration). IPTG was added at 0.5 mM final concentration to start the protein induction and the cells were further grown overnight at 20 °C. After centrifugation, the pellet was stored at −80 °C for at least 4 h to facilitate cell membrane breakage. The frozen pellets were then resuspended in lysis buffer (50 mM potassium phosphate buffer, pH 7.5, 50 mM NaCl, 10% glycerol) containing 1 mM Pefabloc®SC and 5 µL Lysonase TM bioprocessing reagent (Novagen®), agitated for 30 min at RT and sonicated using Ultrasonic Processor. After centrifugation, the cell-free

extract was recovered and stored at −80 °C. Total protein concentrations were determined by the Bradford method with bovine serum albumin as the standard[85]. The samples were analyzed by sodium dodecyl sulfate-polyacrylamide gel electrophoresis (SDS-PAGE) using the Invitrogen NuPAGE system (Supplementary Figs. 5, 6 and 8).

### Purification of enzymes

Purifications were carried out using nickel affinity chromatography either using the Ni-NTA column (QIAGEN), for enzymes selected for amine substrate scope, or in tandem with gel filtration for all the others.

With Ni-NTA column (QIAGEN): the cell-free extracts from 100-mL culture were loaded onto a Ni-NTA column (QIAGEN) according to the supplied protocol. The washing buffer contained 50 mM potassium phosphate buffer (pH 7.5), 50 mM NaCl, 10 % glycerol and 30 mM imidazole. The elution buffer contained 50 mM potassium phosphate buffer (pH 7.5), 50 mM NaCl, 10 % glycerol and 250 mM imidazole. The eluted fractions were desalted using Amicon® Ultra-4 10 K (Merck Millipore®) by three cycles of desalting buffer loading (50 mM potassium phosphate buffer pH 7.5, 50 mM NaCl, 10% glycerol) and centrifugations. Protein concentration of the purified fractions was measured by the Bradford method with bovine serum albumin standard (Bio-Rad®). The purified fractions were also analyzed by SDS-PAGEs using the Invitrogen NuPAGE system (Supplementary Fig. 9). The purified enzymes were stored at −80 °C.

In tandem with gel filtration: the enzymes were purified from a 100-mL culture by nickel affinity chromatography in tandem with gel filtration (Hi Load 16/600 Superdex 200 pg) as described elsewhere[86]. The storage buffer was 50 mM phosphate pH 7.5, 50 mM NaCl, 10% glycerol and 1 mM DTT. Protein concentrations were determined by the Bradford method with bovine serum albumin as the standard. The samples were analyzed by SDS-PAGEs using the Invitrogen NuPAGE system (Supplementary Fig. 10). The purified proteins were stored at −80 °C.

### Amine derivatization protocols and UHPLC-UV conditions

The monitoring of the amine **1b–13b** and **2c-2e** formation was done using a UHPLC-UV method after derivatization with benzoyl chloride (BzCl). The detailed protocol is as followed (in 96-well plates or in Eppendorf tubes 500 μL): to a 20 μL of the reaction mixture were added 50 μL of a 200 mM Na$_2$CO$_3$/NaHCO$_3$ aqueous solution pH 10 and 30 μL of a BzCl solution (7 μL in 1 mL of acetonitrile). The mixture was left at room temperature for 40 min without stirring and then quenched with addition of 20 μL of a 1 M HCl aqueous solution and 30 μL of water/acetonitrile 1/1. After filtration (0.22 μm), the mixture was analyzed by UHPLC-UV (eluent MeCN/H$_2$O 0.1% formic acid A/B; flow 0.5 mL min$^{-1}$; temperature 25 °C; injection volume 3 μL; UV detection at $\lambda$ = 250 nm). The following linear gradients were used. For conditions 1: A/B 20/80 during 1 min, then 20/80 to 70/30 in 3 min (hold 0.5 min), then 70/30 to 20/80 in 1 min and a re-equilibration time of 2 min; for conditions 2: A/B 30/70 during 1 min, then 30/70 to 90/10 in 3.5 min (hold 1 min), then 90/10 to 30/70 in 1 min and a re-equilibration time of 2 min.

The enantiomeric excess was determined by UHPLC-UV analysis after derivatization with 1-fluoro-2,4-dinitrophenyl-5-L-alanine (FDAA). To 20 μL of the reaction mixture were added 8 μL NaHCO$_3$ 1 M (pH 8) and 20 μL of a solution of FDAA prepared in acetone/ethanol 1/1. After incubation at 55 °C for 2 h, the mix was quenched by addition of 4 μL HCl 2 M. After addition of 100 μL MeOH/H$_2$O 1/1, the samples were filtered (0.22 μm) and analyzed by UHPLC-UV (eluent MeOH/H$_2$O 0.1% formic acid A/B; linear gradient A/B 40/60 during 2 min, then 40/60 to 85/15 in 3 min, then 85/15 to 40/60 in 1 min and a re-equilibration time of 3 min; flow 0.3 mL min$^{-1}$; temperature 25 °C; injection volume 3 μL; UV detection at $\lambda$ = 340 nm).

### Activity screening assay with crude cell lysates

Amine-formation assay (UHPLC-UV monitoring): To a reaction mixture (100 μL in 96-well plates), containing 10 mM carbonyl-containing substrate **1a-9a,12a** (with 20% v/v DMSO for **8a**), 0.2 mM NADP$^+$, 0.2 mM NAD$^+$, 3 U mL$^{-1}$ GDH-105, 1.1 eq. glucose in 2 M NH$_4$HCO$_2$/NH$_4$OH buffer (pH 9) (or 250 mM **c** and 200 mM TRIS.HCl pH 9 for reaction with **c**) was added 20 μL of crude cell lysates. Calibration points were prepared using various concentrations of the targeted amine in a mixture containing 2 M NH$_4$HCO$_2$/NH$_4$OH buffer (pH 9) (or 250 mM **c** and 200 mM TRIS.HCl pH 9 for reaction with **c**). Blank reactions were prepared for each enzyme in absence of carbonyl-containing compounds and for each carbonyl substrate with cell-free lysate obtained from the expression of an empty pET22b(+) vector. The reaction mixtures and the calibration points were let at 30 °C for 24 h under agitation at 400 rpm, covered with a pad and a lid. The monitoring of the amine formation was done by UHPLC-UV.

Spectrophotometric screening assay: All the reactions were conducted at 25 °C in 96-microwell plates. Amination reactions: to a reaction mixture (100 μL) containing 10 mM ketone substrate **1a-6a**, 0.5 mM NADH and 0.5 mM NADPH in 2 M NH$_4$HCO$_2$/NH$_4$OH buffer (pH 9) was added 30 μL of cell-free extract. Deamination reactions: to a reaction mixture (100 μL) containing 10 mM amine substrate, 0.5 mM NAD$^+$ and 0.5 mM NADP$^+$ in 100 mM NaHCO$_3$/Na$_2$CO$_3$ buffer (pH 9.8) was added 30 μL of cell-free extract. Absorbance at 340 nm was measured immediately and monitored for 4 h. A background plate was established in the same manner but with a mixture lacking the ketone (amine in the case of deamination reaction) substrate. An active enzyme corresponds to a well exhibiting a higher slope (0-500 s) in the reaction well over the background well.

Globally, 24 h-conversion analysis by UHPLC-UV identified many more hits than the spectrophotometric monitoring based only on the kinetics more subject to background effects. As part of a study aiming at selecting potential valuable biocatalysts among biodiversity, UHPLC-UV monitoring proved to be more suitable.

### Conversion assay with purified enzymes

To a reaction mixture (100 μL in 96-microwell plates), containing 10 mM carbonyl-containing substrate (with 5% v/v DMSO for **7a–13a**), 0.2 mM NADP$^+$, 0.2 mM NAD$^+$, 3 U mL$^{-1}$ GDH-105, 1.1 eq. glucose in 2 M NH$_4$HCO$_2$/NH$_4$OH buffer (pH 9) was added 1 mg mL$^{-1}$ of purified enzymes. In the case of reactions with primary amine [methylamine (**c**), ethylamine (**d**), cyclopropylamine (**e**)], 200 mM TRIS.HCl buffer pH 9 and 250 mM of amine substrate **c-e** were used. Calibration points were prepared using various concentrations of the targeted amine in a mixture containing the reaction buffer (with 250 mM of primary amine for the study of amine spectra), 5% v/v DMSO in the case of corresponding reaction with 5% DMSO, and 20 μL of enzyme purification media. Blank reactions were prepared for each carbonyl substrate in absence of purified enzymes. The reaction mixtures and the calibration points were left at 30 °C for 24 h under agitation at 400 rpm, covered with a pad and a lid. Reactions were performed in duplicates. Amine formation was monitored by UHPLC-UV after BzCl derivatization. The enantiomeric excess was determined by UHPLC-UV analysis after derivatization with FDAA of both racemic and enantiomerically enriched commercially available amines. Calibration points with low amounts of racemic amines determined the detection threshold.

### Molecular docking

**Templates.** The templates used for the docking experiments were homology models generated by the ASMC pipeline (E1QRK4, A0A540X1D9 and MGYP001132756558, A0A229HGK2, MGYP000211951848, A0A365ZD63, MGYP000996099441, MGYP000893205724, A0A4R9C3Q3, IGC-32 and A0A2G6MY80) and the X-ray crystal structure of AmDH4 (PDB ID: 6G1M, chain B) and

*Cfus*AmDH (PDB ID: 6IAU, chain B). The NADP cofactor was added to the templates by copying its coordinates from the *Cfus*AmDH structure.

**Energy minimization.** Previous homology models were energy-minimized using the 'Energy minimization' protocol, within the YASARA Structure software[87,88] (version 22.9.24), which consists of a steepest descent minimization followed by a short simulated annealing in the YASARA NOVA force field[89]. All default settings were used and force field parameters for the ligands (nicotinamide cofactor and substrates) were computed on-the-fly by YASARA.

**Docking with YASARA.** The simulation cell was defined as a 10 Å × 10 Å × 10 Å cubic-shaped box centered on the C4N atom of the nicotinamide moiety. Docking simulations were performed on rigid structures using either the 'dock_run' macro for global docking, or the 'dock_runscreening' for the virtual screening, and the ligand conformations were subsequently analyzed using the "dock_play" macro.

For hypothetical G1 substrates, 2,4-DAP and virtual screening, corresponding ligand structures were downloaded from PubChem[90] in sdf format. Regarding the virtual screening, the set of 1,090 amine-containing molecules was built based on 1) similarity with cyclohexylamine (Tanimoto 50%, MW 45.08-245), 2) similarity with methylbenzylamine (Tanimoto 90%, MW 105.14-245, ROT-BOND 0-5), and 3) substructure matches with methylbenzylamine (MW 119.16-219, ROT-BOND 0-5).

**Docking with AutoDockTool[91].** For docking of 3C-amines, PDB files of amines were generated using Corina[92,93] demo software (https://demos.mn-am.com/corina.html). The simulation cell was defined as a cubic-shape box centered at $x = 25.000$, $y = 28.226$, $z = −0.617$, with dimensions of 46, 52, 54 points (x,y,z) and 0.375 Å spacing. For docking of charged iminium intermediates (amine scope study), SMILES codes were generated using Corina demo software and converted into mol2 format using OpenBabel[94] software. The simulation cell was centered at $x = 30.19$, $y = 28.226$, $z = −0.617$, with dimensions of 34 points (x,y,z) and 0.375 Å spacing. Docking simulations were performed on rigid structures, with no flexibility given to any catalytic pocket residue and the number of Genetic Algorithm (GA) runs was fixed at 10 or 20 using the Lamarckian GA (4.2). Ligand conformations obtained were then analyzed in PyMOL Molecular Graphics System, Version 2.5 Schrödinger, LLC (version 2.5-master-d24468af).

**Active site pocket.** Active site pockets and volumes were computed using CavitOmiX[95] (PyMOL plugin) and visualized with the PyMOL Molecular Graphics System, Version 2.5 Schrödinger, LLC (version 2.5-master-d24468af).

**NetSyn analysis**
To explore the genomic context of the ref-AmDHs, a set of 1,252 UniProt entries, extracted from a non-redundant set of 3,011 ref-AmDH proteins (criteria: 80% of identity over 80% of alignment coverage), was submitted to NetSyn[43]. Among the 1,252 entries, 45 cannot be associated with an ENA identifier or an EMBL file (43 and 2, respectively) and 469 did not have any relevant conserved genomic context (i.e., with a synteny score >=3). Finally, the corresponding network included 738 entries and 72 genomic context clusters generated with the walktrap algorithm (see Supplementary Fig. 4).

**Cofactor specificity study**
A subset of 7224 ref-AmDHs was built by merging UniProt enzymes from the non-redundant set of ref-AmDHs with a previous set of AmDHs for which experimental data are available regarding their preference for NADP/NAD[34]. These protein sequences were aligned using MAFFT[78] (v7.464) and positions were extracted based on the

AmDH4 and *Cfus*AmDH ones (D33-Y38 and D36-R41, respectively). Gap-containing sequences were discarded (248 enzymes) and the remaining ones (6976 enzymes) were included in sequence logos using WebLogo[96] (v3.0).

**Statistics and reproducibility**
No sample size calculation was performed. Given the size of the nat-AmDH family (17,959 sequences), representative enzymes were selected with the support of in silico analysis (sequence identity, comparison of active sites, phylogeny) to cover each family subgroup and reduce the number of experiments to be performed.

Enzyme activity screening was not repeated, except for selected candidates with potential activity for the targeted substrates, for which additional activity assays ($n = 2$) were performed on purified enzymes, as described in the Supplementary Information and Methods.

Regarding the in silico experiments, software parameters are described in the "Methods" section and Supplementary Information to help reproduce the corresponding results.

No randomization was applied to the data. The enzymes collected in this study were assigned to an experimental group on the basis of in silico (sequence identity, active site comparison) and experimental (activity screening) analyses, and by comparing them with data already collected for each AmDH group by Mayol et al.[26].

Only recombinant proteins and *E. coli* cells were involved in this study (no animal or human participants).

Data collection based on genomic criteria was blind, as we searched for any NAD(P)-dependent enzyme, regardless of the enzymatic reaction performed. However, the updating of the AmDH family, the selection of specific enzymes within this family and the selection of substrates were not carried out blindly, as the AmDH features were necessary to set up the in silico and in vitro experiments described in the "Methods" section.

**Reporting summary**
Further information on research design is available in the Nature Portfolio Reporting Summary linked to this article.

## Data availability
Data generated in this study can be accessed through the Zenodo repository (https://doi.org/10.5281/zenodo.7889419). It contains libraries of NAD(P)-dependent enzyme sequences and ref-AmDH sequences, HMM libraries of NAD(P)-dependent protein subfamilies and nat-AmDHs, ref-AmDH homology models, as well as sequences of representative ref-AmDHs tested and of heterologously expressed nat-AmDHs with specific feature. PDB accessions were obtained from RCSB PDB [https://www.rcsb.org/] and include 6G1M, 6IAU, 6IAQ, and 7ZBO. Protein sequences were extracted from the genomic and metagenomic databases listed in Supplementary Table 1. All data supporting the findings of this study are available within the paper and its Supplementary Information and Data. Source data are provided with this paper.

## Code availability
The ASMC code was previously described in ref. 41. and is now freely available at https://doi.org/10.5281/zenodo.10979029.

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

## Acknowledgements

The authors thank A. Debard for the enzyme production and V. Pellouin for the spectrophotometric enzymatic assay. They also thank A. Perret and P. Sirvain for large-scale purification of the described enzymes and E. Soumar for his contribution in the amine substrate scope study. This work was supported by the Agence Nationale de la Recherche (ANR) through the MODAMDH (ANR-19-CE07-0007) and ALADIN (ANR-21-ESRE-0021) projects, and by Commissariat à l'énergie atomique et aux énergies alternatives (CEA), the CNRS and the University of Evry Val d'Essonne—University of Paris-Saclay.

## Author contributions

C.V.V. and D.V. conceived the project. E.E., D.V. and C.V.V. managed it with the input of A.Z., V.d.B and G.G. All the in silico work was conducted by E.E. with the support of D.V., R.M., E.P., M.S., K.B., L.D. and C.V.V. J.-L.P., E.E. and L.D. performed the enzyme selection for in vitro analysis. J.-L.P. carried out the gene cloning, protein expression and purification on a small scale. L.D., C.V.V. and A.F.-J. performed the in vitro experiments. E.E., L.D., D.V. and C.V.V. wrote the manuscript with input from R.M., K.B., G.G., J.-L.P., M.S., A.Z. and V.d.B.

## Competing interests

The authors declare no competing interests.
