## [Peer Review File · Nature Communications]

REVIEWER COMMENTS

Reviewer #1 (Remarks to the Author):

In this manuscript, Elisée et al. present a comprehensive workflow of functional annotation and characterisation of Amine Dehydrogenases, leveraging the extensive new sequence information derived from metagenomics experiments. Their work is very detailed and combines computational identification and clustering of native AmDHs, as well as experimental validation of a significant fraction of the identified hits.

Although the paper is in scope for Nature Communications and would be useful to a broad audience of both biochemists/enzymologists and bioinformaticians, it is rather long, with critical findings often being diluted in excessive reporting of data. In general, especially at the experimental part, reading was tedious, while it felt that results and methods are not clearly distinguished.

In light of these broad observations and the following comments, I would invite the authors to re-structure and/or re-write the manuscript, so content can be followed more easily.

1) I would propose re-writing according to Figure 1, i.e. have sections 'Environmental sampling', 'Clustering' and 'in vitro experiments' where the processes are discussed (briefly). Then include a Results and Discussion section where the observations are considered.

2) Introduction can be more concise, focusing more on the current limitations and needs in AmDH functional annotation, and in the authors' contribution to this.

3) Section "Extending the reference nat-AmDH family":

a. It is not clear if G1-G5 are an established classification for native AmDHs or a nomenclature based on the authors' phylogenetic analysis.

b. Out of the extended occupancies in these five groups, how many of the hits are expected to be non-functional (i.e. pseudoenzymes). Active site analyses by ASMC should provide enough insight for

functionally critical catalytic residues, whose mutation would be deleterious for function. Maybe also look at variant effects in Clinvar?

4) Section “Searching for nat-AmDH distant homologs”:

a. The search is purely sequence based, though enzymes can diverge so much that homology can only be inferred by structure and in some cases only by the structure of the catalytic core (e.g. in some phosphatases, where divergence is so extensive it led to fold shift). Authors should at least discuss structure based ways of capturing homology – 3D template search is one of method.

b. It would also be interesting to see which CATH superfamilies are covered. This can be done using experimental structures where available or predicted models (AF2 or Meta predictions).

c. Fig 2C: How was this subset of sequences used to draw the matrix selected? If it is random sampling, cluster sizes should reflect the original sizes seen in the trees. This should either be explained or removed, since the diversity message is already given from the tree representation in Figs 2A.

5) ASMC is a useful pipeline. I believe the authors should make their code publicly available in a scalable and reproducible format.

6) Fig 3A: Catalytic residues should be highlighted on the active site local sequence and colouring of sequence alignment should be explained in the legend.

7) Section “Overview of the biocatalytic activity of the AmDH family”: Again there is too much detail and confusion of data with observations. The same applies to Supplementary information. Also last sentence of the section should be rewritten since it is difficult to understand – maybe the authors meant “inferred” instead of “obtained”?

8) Fig 4: 1b-5b should be 1a-5a? Also, for consistency, it is better to use phylogenetic classification for all examples rather than a mixture of active site clustering and whole-sequence phylogeny. Given that according to authors' findings Fig 3B, active site similarity AmDHs is correlated with overall sequence identity, so better use overall sequence identity for everything and then add G1-G5 labels.

9) Section "Selection of enzymes with unprecedented substrate scope":

a. Very interesting and novel findings, useful for enzymologists, but too long and diluted text. Should be more succinct.

b. Phe140 position in Fig 6. What is the effect of His->Phe and Phe->Ala mutations? It is not clear if they affect the pocket size and physical chemical environment. Otherwise this is a good analysis.

10) Labels in all figures should be enlarged to be clearer on screen. Titles should also be added on plots like Figs 2A-B.

11) Section "Cofactor specificity of nat-AmDHs": It would be interesting (more as a suggestion than a revision) to see if there is any evidence for absence of cofactors in ancestral homologues. Have the authors considered this in the context of evolution of enzymes and cofactors?

12) Minor grammar errors scattered throughout the text.

Reviewer #2 (Remarks to the Author):

Overall, this is an excellent work in terms of significance and potential impact in the field of biocatalysis, and discovery and characterization of new enzymes.

The immediate impact is that this work greatly extends our knowledge in: i) understanding the biodiversity of amine dehydrogenases (AmDH)s; ii) allows us to draw better correlations of enzyme structure/sequence/cluster vs. catalytic activity/substrate scope; iii) provides interesting new scaffolds of AmDHs for further enzyme engineering; and iv) along with previous publications, it shows that the AmDH activity can be expanded beyond ammonia as amine donor.

I point out another (wider) element of novelty that the authors could probably try to emphasize a bit more. I think that the workflow used for the discovery, classification, and investigation of these AmDHs could be applied on other enzyme families. Can the authors comment on that in a revised manuscript? What about the broader applicability of the methods used in this manuscript for the discovery of other dehydrogenases or other oxidoreductases or enzymes from other EC classes?

The manuscript is in general well written. Some minor editing will be needed. The conclusion and the discussion are fully supported by the experimental data for what concerns the workflow of bioinformatics to discover and classify the new amine dehydrogenases and their respective sequences. The main issues that I found in this manuscript are related to place the results obtained in terms of substrate scope and catalytic activity within the state-of-the-art in the literature. Many relevant previous publications were not included in the discussion. The author should include these papers and modify the discussion accordingly. Some statements in the manuscript are currently wrong or misleading. These points for improvements are explained in detail in the second part of this report.

All the experiments were carefully designed and performed. The manuscript and SI contain all the information that allows for the reproducibility of the work. The experimental part is comprehensive, yet, concise. All the additional information, tables and figures are in the SI. The quality of the illustrations is excellent for design and clarity. I had only one remark regarding Figure 1 (see later).

In conclusion, I think that this manuscript can be suitable for publication in Nature Communication, providing that the points below will be addressed in a revision:

1) Some of the references [1–4] could have been selected better. There are some interesting reviews about biocatalysis in the context of green chemistry that are appropriate. I suggest making a survey on the recently published reviews on this topic and ponder a selection (the number of references could be extended).

Some possible examples are these reviews that provide some perspectives on the future of biocatalysis:

a) France, S.P., Lewis, R.D. and Martinez, C.A. (2023) The Evolving Nature of Biocatalysis in Pharmaceutical Research and Development. *JACS Au* 3 (3), 715-735.

b) Hauer, B., Embracing Nature's Catalysts: A Viewpoint on the Future of Biocatalysis. ACS Catal. 2020, 10 (15), 8418-8427

c) Sheldon, R. A.; Woodley, J. M., Role of Biocatalysis in Sustainable Chemistry. Chem. Rev. 2018, 118 (2), 801-838.

A recently published perspective in this area is available at:

d) Buller, R., Lutz, S., Kazlauskas, R.J., Snajdrova, R., Moore, J.C., Bornscheuer, U.T. (2023), From nature to industry: harnessing enzymes for biocatalytic processes, Science, 382, eadh8615.

2) The definition "green industry" should be changed into "green chemical industry" in the introduction.

3) References [9,10] could also have been selected better. There are nice reviews or book chapters on biocatalytic synthesis of chiral amines. In this context, ref. [9] from 2009 is a bit outdated. Again, I recommend searching in the recent literature and make a better selection.

4) There some issues in the cited papers in the following paragraph:" This asymmetric reductive amination of ketones can be accomplished using engineered Amine Dehydrogenases (eng-AmDHs) from wild-type amino acid dehydrogenases (aaDHs), native AmDHs (nat-AmDHs) [cit. 13] recently identified by our group, some reductive aminases (RedAms), a subclass of imine reductases (IREDs) active with ammonia as amine source, and an engineered ϵ -deaminating L-lysine dehydrogenase [cit. 14-17].

When directly introducing these enzyme classes, the first report for each of them (at least) must always be cited and not just three general reviews [cit. 14-17]. The authors only cited their own original contribution regarding "native AmDHs (nat-AmDHs)" at cit. 13. However, all the ones below for the other cases were omitted:

When mentioning "wild-type amino acid dehydrogenases (aaDHs)", some of the works by Bommaris' group should be cited, at least the first report on amine dehydrogenase:

Abrahamson, M. J.; Vazquez-Figueroa, E.; Woodall, N. B.; Moore, J. C.; Bommarius, A. S., Development of an Amine Dehydrogenase for Synthesis of Chiral Amines. *Angew. Chem. Int. Ed.* 2012, 51 (16), 3969-3972.

When mentioning 'some reductive aminases (RedAms), a subclass of imine reductases (IREDs) active with ammonia as amine source', the first report by Turner's group should be cited:

Aleku, G. A.; France, S. P.; Man, H.; Mangas-Sanchez, J.; Montgomery, S. L.; Sharma, M.; Leipold, F.; Hussain, S.; Grogan, G.; Turner, N. J., A reductive aminase from *Aspergillus oryzae*. *Nat. Chem.* 2017, 9 (10), 961-969.

When mentioning "and an engineered ϵ -deaminating L-lysine dehydrogenase", the first report by Mutti's group should be cited:

Tseliou, V.; Knaus, T.; Masman, M. F.; Corrado, M. L.; Mutti, F. G., Generation of amine dehydrogenases with increased catalytic performance and substrate scope from epsilon-deaminating L-Lysine dehydrogenase. *Nat. Commun.* 2019, 10 (1), 3717.

Note that these are just three evident examples that came quickly to my view. I think it would be desirable (and fair) to cite a few more original papers by other groups. In contrast, I do not see a great value in citing in this section all the three general reviews [cit. 15-17].

5) Figure 1 could probably be improved and made clearer. The point for possible confusion is that the figure divides the workflow in three steps: environmental sampling, clustering & selection, and in vitro experiments. In contrast, the figure caption has five steps. I suggest modifying either the figure caption or the figure itself to have the same number of steps and a clear indication of the steps involved.

6) The reaction described in Table 1, Table 2, Figure 7, and discussed in the text were performed at quite low substrate concentration (10 mM). Did the author try at higher substrate concentrations like 50 mM or 100 mM? A comment on the effect of substrate concentration would be valuable.

7) Table 1, footnote reports: “Uncertainties represent standard deviations of two independent experiments”. A standard deviation based on two sample is not properly statistically defined. I suggest changing the text with e.g., “Uncertainties represent the range of values obtained with two independent experiments” or something like that.

8) The expression “larger substrates” could be substituted with “bulkier” or “more sterically demanding” substrates.

9) It is reported for some of the new enzymes that “Ala151 (P5) opens the cavity (...)” and “P5 appears to be the critical residue to accommodate larger substrates; this hypothesis could not be verified in CfusAmDH as the P5-mutant is unstable”.

As a suggestion: one option could be to take one of the “Ala151” stable enzymes and try to mutate the position 151 with a glycine. If the mutation is tolerated, even bulkier substrates should be in principle accommodated.

10) Consider the sentence: “The ketones bearing the carbonyl function at the third carbon atom of the chain length (3C-ketones) was another class of target substrates. Even now, these are still poorly described in reductive amination even by other classes of enzymes such as RedAms, IREDs or eng-AmDHs, the only reported examples leading to the formation of the (R)-enantiomer with low enantiomeric excesses [cit. 38]”

This is not correct. This type of more bulky-bulky ketones for amination have been described for AmDHs (producing the R-enantiomer) in a few more other publications:

a) Knaus, T.; Böhmer, W.; Mutti, F. G., Amine dehydrogenases: efficient biocatalysts for the reductive amination of carbonyl compounds. *Green Chem.* 2017, 19, 453-463.

b) Wang, D.-H.; Chen, Q.; Yin, S.-N.; Ding, X.-W.; Zheng, Y.-C.; Zhang, Z.; Zhang, Y.-H.; Chen, F.-F.; Xu, J.-H.; Zheng, G.-W., Asymmetric Reductive Amination of Structurally Diverse Ketones with Ammonia Using a Spectrum-Extended Amine Dehydrogenase. *ACS Catal.* 2021, 11 (22), 14274-14283.

Furthermore, amination of cyclic bulky-bulky ketones with AmDHs have been described here:

c) Lv, T.; Feng, J.; Chen, X.; Luo, Y.; Wu, Q.; Zhu, D.; Ma, Y., Desymmetric Reductive Amination of 1,3-Cyclopentadiones to Single Stereoisomer of β -Amino Ketones with an All-Carbon Quaternary Stereocenter by Engineered Amine Dehydrogenases. *ACS Catal.* 2023, 13 (7), 5053-5061.

Also, of possible interest for the discussion in the manuscript, are substrates with an OH in position 1, but still having similar structural feature (OH-CH₂-CO-R):

d) Ming, H.; Yuan, B.; Qu, G.; Sun, Z., Engineering the activity of amine dehydrogenase in the asymmetric reductive amination of hydroxyl ketones. *Catal. Sci. Technol.* 2022, 12 (19), 5952-5960.

e) Chen, F.-F.; Cosgrove, S. C.; Birmingham, W. R.; Mangas-Sanchez, J.; Citoler, J.; Thompson, M. P.; Zheng, G.-W.; Xu, J.-H.; Turner, N. J., Enantioselective Synthesis of Chiral Vicinal Amino Alcohols Using Amine Dehydrogenases. *ACS Catal.* 2019, 9 (12), 11813-11818.

I think that the paragraph should be modified and put into the perspective of the literature above that should also be cited in this regard (at least suggestions from a to c).

Finally, the authors correctly emphasize that they produce the opposite enantiomer (S) as the point of novelty. This was properly stated: "To the best of our knowledge, these enzymes are the only ones yet reported to provide the (S)-enantiomer of such 3C ketones (...)" This sentence could probably be moved within the previous paragraph to make clear to the reader this element of novelty earlier in the text.

11) Regarding the amination of ketones with amine donors larger than ammonia, I could not find the absolute configuration of the formed amine products (neither in main manuscript nor in Supplementary Figure 25). Were they (S)-configured?

12) Regarding the amination of ketones with amine donors larger than ammonia using AmDHs, there is another publication prior the cited ref. 40 (NfRedAm or AdRedAm) that is:

Tseliou, V.; Masman, M. F.; Böhmer, W.; Knaus, T.; Mutti, F. G., Mechanistic Insight into the Catalytic Promiscuity of Amine Dehydrogenases: Asymmetric Synthesis of Secondary and Primary Amines. *ChemBioChem* 2019, 20 (6), 800-812.

This work could also be placed in the context of this discussion besides ref. 40. Here, the amines were enantioenriched (R). I assume that the authors got the (S) with their enzymes. This feature could be compared and highlighted.

13) For the reasons explained previously, the sentence in the conclusion: “Moreover, activity towards C3-ketones, rarely reported for biocatalysts performing reductive amination, was found to be (...)” should be modified.

14) The next sentence in the conclusion should also be modified and placed better within the state-of-the-art: “(...) moderate to high activity for both alkylamine (methylamine and cyclopropylamine) and ammonia was verified for some enzymes, overcoming the restriction to primary amines and giving access to secondary amines”.

15) Supplementary Figure 5: please, indicate the band of the expressed enzyme (e.g., red circle) on the gel. The bands of the expressed enzymes are not well visible due to the moderate expression level. This could also be done for the other two gels in SI fig. 6 and 12 although here some bands are a lot more visible already (but others not).

Reviewer #3 (Remarks to the Author):

This manuscript describes an expansion of the native amine hydrogenase (nat-AmDH) family from 2,011 sequences (presumably in the UniProt database in 2020?) to 17,959 sequences from ten genome and

metagenome databases (from 2019, 2020, 2021). The expanded family was then subjected to screening of representative members to provide a description of the substrate specificities; not surprisingly, the authors discovered novel specificities that would be useful in using members of the family to provide access to novel chiral amines.

The expanded membership was discovered using a new HMM for the C-terminal catalytic domain sequences of the original set of sequences. The sequences for the N-terminal Rossmann-fold domain were excluded to eliminate false positives, although the sequences identified using the new HMM for the catalytic domain were required to contain an N-terminal Rossmann-fold domain. This strategy is, in the opinion of this reviewer, the most generally instructive part of the manuscript.

Despite the length of the manuscript, many of the details of this expansion strategy are difficult to find. Most basic, the authors do not state where/how the original 2,011 sequences were identified? UniProt? Which Pfam or InterPro family?

And, it is somewhat frustrating that the databases that were used are quite dated--2019, 2020, 2021. As a result, the number of sequences is now much larger?

This manuscript does not mention AlphaFold structures. I suspect this is not surprising, given that the downstream enzymological characterization work presumably required significant time and effort. But, readers now would expect that a manuscript describing the expansion of a Pfam-curated family (?) would involve only sequence (HMM) methods. This strategy likely will not identify more divergent members of the family that would be expected to have additional novel substrate specificities.

Point-by-point response to the Reviewers' comments

Reviewer #1 (Remarks to the Author):

In this manuscript, Eliséé et al. present a comprehensive workflow of functional annotation and characterisation of Amine Dehydrogenases, leveraging the extensive new sequence information derived from metagenomics experiments. Their work is very detailed and combines computational identification and clustering of native AmDHs, as well as experimental validation of a significant fraction of the identified hits.

Although the paper is in scope for Nature Communications and would be useful to a broad audience of both biochemists/enzymologists and bioinformaticians, it is rather long, with critical findings often being diluted in excessive reporting of data. In general, especially at the experimental part, reading was tedious, while it felt that results and methods are not clearly distinguished.

In light of these broad observations and the following comments, I would invite the authors to re-structure and/or re-write the manuscript, so content can be followed more easily.

The paragraphs "Overview of the biocatalytic activity of the AmDH family" and "Selection of enzymes with unprecedented substrate scope" (renamed Structure-based selection of enzymes with unprecedented substrate scope") dealing with experimental results and related discussions, have been re-written to be more concise. Some sentences have been moved to "Methods" to distinguish between results and methods and to avoid redundancy of information. Others have been removed to avoid the dilution of critical findings and ease the reading of the manuscript.

1) I would propose re-writing according to Figure 1, i.e. have sections 'Environmental sampling', 'Clustering' and 'in vitro experiments' where the processes are discussed (briefly). Then include a Results and Discussion section where the observations are considered.

The subsections in 'Results and Discussion' section have been modified to align more clearly with Figure 1. This section is now divided into 1) 'Environmental sampling, clustering and analysis', 2) 'Structural analysis of the ref-AmDH active sites', 3) 'In vitro experiments: selection and enzymatic tests', which includes the previous sub-sections 'Overview of the biocatalytic activity of the AmDH family' and the renamed 'Structure-based selection of enzymes with unprecedented substrate scope'. We have moved the sentences describing more of the processes to the 'Methods' Section, and rewritten the paragraphs to be more concise as suggested. The subsections in the Methods section have been maintained to assist the reader, with each step separately described.

2) Introduction can be more concise, focusing more on the current limitations and needs in AmDH functional annotation, and in the authors' contribution to this.

We have slightly reduced the introduction while retaining the general introduction on biocatalysis as we think this is useful for the general scientific readership of this journal. We

have also retained sentences relating biocatalysed amine synthesis and the data already published on nat-AmDHs to satisfy the various references and also to address the points raised by Reviewer 2.

3) Section “Extending the reference nat-AmDH family”:

a. It is not clear if G1-G5 are an established classification for native AmDHs or a nomenclature based on the authors’ phylogenetic analysis.

G1-G5 is an established classification for native AmDHs that was reported in Mayol et al. 2019. This point has been clarified in the main text: “Figure 2B highlights the extension of the previous G1-G5 groups of the nat-AmDH family”.

b. Out of the extended occupancies in these five groups, how many of the hits are expected to be non-functional (i.e. pseudoenzymes). Active site analyses by ASMC should provide enough insight for functionally critical catalytic residues, whose mutation would be deleterious for function. Maybe also look at variant effects in Clinvar?

If we consider only the glutamate residue P3 as essential for a hit to be functional, 706 out of 9,886 ASMC hits should be non-functional – these are mainly present in the G5 group.

Regarding the variant effects, ClinVar appears to provide access to established relationships between human variants and observed health status, which may not be appropriate in the case of AmDHs, since they are not present in the human genome.

4) Section “Searching for nat-AmDH distant homologs”:

a. The search is purely sequence based, though enzymes can diverge so much that homology can only be inferred by structure and in some cases only by the structure of the catalytic core (e.g. in some phosphatases, where divergence is so extensive it led to fold shift). Authors should at least discuss structure based ways of capturing homology – 3D template search is one of method.

We thank the reviewer for this suggestion. We also tried to search for active site patterns using a geometric method (catalophore) as proposed by Steinkellner et al., but without success. We found numerous limitations to this method in our case, which are discussed at the end of the section “Environmental sampling, clustering and analysis”.

b. It would also be interesting to see which CATH superfamilies are covered. This can be done using experimental structures where available or predicted models (AF2 or Meta predictions).

Not surprisingly, the main CATH annotation for the AmDH family is for the Rossmann-like NAD(P)-binding domain (CATH ID: 3.40.50.720 - version 4.3.0), which corresponds to the N-terminal domain of AmDH. No CATH annotation was found for the C-terminal domain of AmDH.

c. Fig 2C: How was this subset of sequences used to draw the matrix selected? If it is random sampling, cluster sizes should reflect the original sizes seen in the trees. This should either be explained or removed, since the diversity message is already given from the tree representation in Figs 2A.

We considered all members of the G1-G4 groups. For clarity, only 500 out of 4,191 members of the G5 group were randomly selected. We have removed Figure 2C from the main text as the diversity is already highlighted by the tree representation in Figure 2A. Nevertheless, Figure 2C has been moved to the SI (as Supplementary Figure 2) as the diversity expressed through sequence identity numbers also constitutes useful data when studying enzymes for the purposes of biocatalysis.

5) ASMC is a useful pipeline. I believe the authors should make their code publicly available in a scalable and reproducible format.

The ASMC code (version 2016 used here) is now freely available on GitHub (in a scalable and reproducible format as far as possible). A renewed version of the ASMC pipeline should be published later in 2024.

6) Fig 3A: Catalytic residues should be highlighted on the active site local sequence and colouring of sequence alignment should be explained in the legend.

The Figure and caption have been reworked.

7) Section “Overview of the biocatalytic activity of the AmDH family”: Again there is too much detail and confusion of data with observations. The same applies to Supplementary information. Also last sentence of the section should be rewritten since it is difficult to understand – maybe the authors meant “inferred” instead of “obtained”?

See point 1) : The relevant paragraph has been simplified by removing some sentences. The sentence highlighted has been re-written for clarification: “Such wide sequence homology could not have been obtained by protein engineering, thus emphasizing the benefit of this type of workflow.”

8) Fig 4: 1b-5b should be 1a-5a? Also, for consistency, it is better to use phylogenetic classification for all examples rather than a mixture of active site clustering and whole-sequence phylogeny. Given that according to authors’ findings Fig 3B, active site similarity AmDHs is correlated with overall sequence identity, so better use overall sequence identity for everything and then add G1-G5 labels.

Figure 4 has been modified using the phylogenetic classification. The label 1a-5a, referring to ketones has been retained in the Figure to clearly illustrate that these enzymes were tested against various substrates. The legend has been clarified as follows: ‘Analytical yields in **1b-5b** from tested substrates **1a-5a** are...’.

9) Section “Selection of enzymes with unprecedented substrate scope”:

a. Very interesting and novel findings, useful for enzymologists, but too long and diluted text. Should be more succinct.

Some sentences that refer more to the state-of-the-art have been moved with more concise phrasing to the Introduction and others have been re-written to generate a more succinct paragraph.

b. Phe140 position in Fig 6. What is the effect of His->Phe and Phe->Ala mutations? It is not clear if they affect the pocket size and physical chemical environment. Otherwise this is a good analysis.

1) The Phe140->Ala161 mutation leads to enlargement of the active site, freeing up the space occupied by the Phe side chain (in minor relation to the displacement of Trp156 relative to Trp145).

2) Regarding the Phe->His mutation:

- a. in A0A229HGK2 (middle), the Phe140->His145 and Leu177->Val182 mutations enable a slight widening of the active site compared with CfusAmDH, but the Thr166->Leu171 mutation still restricts the active site.
- b. in MGYP000211951848 (right), the Phe140->His135 and Thr166->Ala161 mutations led to a real enlargement of the active site.
- c. could affect the physical chemical environment (His being a basic residue)

10) Labels in all figures should be enlarged to be clearer on screen. Titles should also be added on plots like Figs 2A-B.

Figures have been modified according to this recommendation, especially Figure 4, Figure 6 and Supplementary Figure 1. In addition, former Supplementary Figures 4, 7, 9 and 15 providing *in vitro* detailed screening data have been moved to the Supplementary Data as Excel files to facilitate the reading and sharing of these data. The database from which each sequence comes has been included in these Excel sheets.

11) Section "Cofactor specificity of nat-AmDHs": It would be interesting (more as a suggestion than a revision) to see if there is any evidence for absence of cofactors in ancestral homologues. Have the authors considered this in the context of evolution of enzymes and cofactors?

We didn't consider the absence of cofactors in ancestral homologues. However, thanks to the new Pfam entry PF19328 / IPR045760 annotating the AmDH catalytic domain, we can retrieve 24 domain architectures, from InterPro, in which this C-terminus domain is coupled to a N-terminus annotation (Pfam or SCOP Superfamily SSF51735) related to the NAD(P)-binding domain superfamily.

12) Minor grammar errors scattered throughout the text.

Grammar errors have been corrected.

Reviewer #2 (Remarks to the Author):

Overall, this is an excellent work in terms of significance and potential impact in the field of biocatalysis, and discovery and characterization of new enzymes.

The immediate impact is that this work greatly extends our knowledge in: i) understanding the biodiversity of amine dehydrogenases (AmDH)s; ii) allows us to draw better correlations of enzyme structure/sequence/cluster vs. catalytic activity/substrate scope; iii) provides interesting new scaffolds of AmDHs for further enzyme engineering; and iv) along with previous publications, it shows that the AmDH activity can be expanded beyond ammonia as amine donor.

I point out another (wider) element of novelty that the authors could probably try to emphasize a bit more. I think that the workflow used for the discovery, classification, and investigation of these AmDHs could be applied on other enzyme families. Can the authors comment on that in a revised manuscript? What about the broader applicability of the methods used in this manuscript for the discovery of other dehydrogenases or other oxidoreductases or enzymes from other EC classes?

Yes, that was the ultimate goal when we started the project. This broader applicability was emphasized at the end of the revised manuscript with the sentence "This workflow can be used directly for other NADP-dependent oxidoreductases benefiting from already retrieving NADP-dependent enzymes from (meta)genomics databases, and we are currently in the process of generalizing it to allow its applicability to other families of enzymes."

The manuscript is in general well written. Some minor editing will be needed. The conclusion and the discussion are fully supported by the experimental data for what concerns the workflow of bioinformatics to discover and classify the new amine dehydrogenases and their respective sequences. The main issues that I found in this manuscript are related to place the results obtained in terms of substrate scope and catalytic activity within the state-of-the-art in the literature. Many relevant previous publications were not included in the discussion. The author should include these papers and modify the discussion accordingly. Some statements in the manuscript are currently wrong or misleading. These points for improvements are explained in detail in the second part of this report.

All the experiments were carefully designed and performed. The manuscript and SI contain all the information that allows for the reproducibility of the work. The experimental part is comprehensive, yet, concise. All the additional information, tables and figures are in the SI. The quality of the illustrations is excellent for design and clarity. I had only one remark regarding Figure 1 (see later).

In conclusion, I think that this manuscript can be suitable for publication in Nature Communication, providing that the points below will be addressed in a revision:

1) Some of the references [1–4] could have been selected better. There are some interesting reviews about biocatalysis in the context of green chemistry that are appropriate. I suggest making a survey on the recently published reviews on this topic and ponder a selection (the number of references could be extended).

Some possible examples are these reviews that provide some perspectives on the future of biocatalysis:

a) France, S.P., Lewis, R.D. and Martinez, C.A. (2023) The Evolving Nature of Biocatalysis in Pharmaceutical Research and Development. *JACS Au* 3 (3), 715-735.

b) Hauer, B., Embracing Nature's Catalysts: A Viewpoint on the Future of Biocatalysis. *ACS Catal.* 2020, 10 (15), 8418-8427

c) Sheldon, R. A.; Woodley, J. M., Role of Biocatalysis in Sustainable Chemistry. *Chem. Rev.* 2018, 118 (2), 801-838.

A recently published perspective in this area is available at:

d) Buller, R., Lutz, S., Kazlauskas, R.J., Snajdrova, R., Moore, J.C., Bornscheuer, U.T. (2023), From nature to industry: harnessing enzymes for biocatalytic processes, *Science*, 382, eadh8615.

The previous reference 1 (2018) has been removed, and proposed references a), b) and d) have been added, in addition to the recent Reviews/Perspective:

- Sheldon, A., Brady, D. (2022) Green Chemistry, Biocatalysis, and the Chemical Industry of the Future, *ChemSusChem*, 15, 9, e202102628
- Lozano, P. García-Verdugo, E. (2023) *Green Chem*, From green to circular chemistry paved by biocatalysis, 25, 7041-7057.
- Bryan M.C. et al. (2022) Green Chemistry Articles of Interest to the Pharmaceutical Industry,, *Org. Process Res. Dev.*, 26, 2, 251–262
- Buller et al (2023), From nature to industry: Harnessing enzymes for biocatalysis, *Science*, 382, 899; [10.1126/science.adh8615](https://doi.org/10.1126/science.adh8615).

2) The definition “green industry” should be changed into “green chemical industry” in the introduction.

Corrected

3) References [9,10] could also have been selected better. There are nice reviews or book chapters on biocatalytic synthesis of chiral amines. In this context, ref. [9] from 2009 is a bit outdated. Again, I recommend searching in the recent literature and make a better selection.

The previous reference 9 has been removed and replaced by:

- Mutti, F., Knaus, T. (2021), Enzymes Applied to the Synthesis of Amines, Chapter 6 in “Biocatalysis for Practitioners: Techniques, Reactions and Applications”. <https://doi.org/10.1002/9783527824465.ch6>
- Sangster, J.J. et al (2021) New Trends and Future Opportunities in the Enzymatic Formation of C–C, C–N, and C–O bonds, *ChemBioChem*, 23, 6, e202100464
- Grogan, G. (2018) Synthesis of chiral amines using redox biocatalysis, *Curr. Opin Chem Biol* 10.1016/j.cbpa.2017.09.008

4) There some issues in the cited papers in the following paragraph.” This asymmetric reductive amination of ketones can be accomplished using engineered Amine Dehydrogenases (eng-AmDHs) from wild-type amino acid dehydrogenases (aaDHs), native

AmDHs (nat-AmDHs) [cit. 13] recently identified by our group, some reductive aminases (RedAms), a subclass of imine reductases (IREDs) active with ammonia as amine source, and an engineered ϵ -deaminating L-lysine dehydrogenase [cit. 14-17].

When directly introducing these enzyme classes, the first report for each of them (at least) must always be cited and not just three general reviews [cit. 14-17]. The authors only cited their own original contribution regarding “native AmDHs (nat-AmDHs)” at cit. 13. However, all the ones below for the other cases were omitted:

When mentioning “wild-type amino acid dehydrogenases (aaDHs)”, some of the works by Bommarius’ group should be cited, at least the first report on amine dehydrogenase:

Abrahamson, M. J.; Vazquez-Figueroa, E.; Woodall, N. B.; Moore, J. C.; Bommarius, A. S., Development of an Amine Dehydrogenase for Synthesis of Chiral Amines. *Angew. Chem. Int. Ed.* 2012, 51 (16), 3969-3972.

When mentioning “some reductive aminases (RedAms), a subclass of imine reductases (IREDs) active with ammonia as amine source”, the first report by Turner’s group should be cited:

Aleku, G. A.; France, S. P.; Man, H.; Mangas-Sanchez, J.; Montgomery, S. L.; Sharma, M.; Leipold, F.; Hussain, S.; Grogan, G.; Turner, N. J., A reductive aminase from *Aspergillus oryzae*. *Nat. Chem.* 2017, 9 (10), 961-969.

When mentioning “and an engineered ϵ -deaminating L-lysine dehydrogenase”, the first report by Mutti’s group should be cited:

Tseliou, V.; Knaus, T.; Masman, M. F.; Corrado, M. L.; Mutti, F. G., Generation of amine dehydrogenases with increased catalytic performance and substrate scope from epsilon-deaminating L-Lysine dehydrogenase. *Nat. Commun.* 2019, 10 (1), 3717.

Note that these are just three evident examples that came quickly to my view. I think it would be desirable (and fair) to cite a few more original papers by other groups. In contrast, I do not see a great value in citing in this section all the three general reviews [cit. 15-17].

The references of each original work have been added, in addition to two references which directly followed the first publication of Bommarius and co-workers (<https://chemistry-europe.onlinelibrary.wiley.com/doi/abs/10.1002/cctc.201902364>) and Turner and co-workers (<https://pubs.rsc.org/en/content/articlelanding/2020/sc/d0sc02253e>) because they gave rise to all of the following ones for generalization on other enzymes and examples of application in synthesis. The reviews (previously [14-17]) have been retained because they bring together all of the work that resulted from these initial results. Also, another very recent review on NADP-dependent enzymes performing reductive amination by Yuan et al (DOI: 10.1039/d3cs00391d) has been added.

5) Figure 1 could probably be improved and made clearer. The point for possible confusion is that the figure divides the workflow in three steps: environmental sampling, clustering & selection, and in vitro experiments. In contrast, the figure caption has five steps. I suggest

modifying either the figure caption of the figure itself to have the same number of steps and a clear indication of the steps involved.

The Figure and caption have been modified accordingly, also taking into account the suggestion of Reviewer 1 to align these steps with the main text subsections.

6) The reaction described in Table 1, Table 2, Figure 7, and discussed in the text were performed at quite low substrate concentration (10 mM). Did the author try at higher substrate concentrations like 50 mM or 100 mM? A comment on the effect of substrate concentration would be valuable.

The *in vitro* experiments were performed to demonstrate the catalytic reductive amination activity of the selected enzymes and to gain a first insight into the extended substrate spectrum of these enzymes. The goal here was not to fully describe these newly identified enzymes, which would require extensive further work outside the scope of this study. Some of them are currently being further investigated for biocatalytic applications (including increasing substrate loadings) and characterized biochemically. These results will be detailed in a future publication.

7) Table 1, footnote reports: "Uncertainties represent standard deviations of two independent experiments". A standard deviation based on two samples is not properly statistically defined. I suggest changing the text with e.g., "Uncertainties represent the range of values obtained with two independent experiments" or something like that.

We agree with this comment. This has been corrected in Table 1 and 2 and Figure 7.

8) The expression "larger substrates" could be substituted with "bulkier" or "more sterically demanding" substrates.

This has been corrected throughout the manuscript.

9) It is reported for some of the new enzymes that "Ala151 (P5) opens the cavity (...)" and "P5 appears to be the critical residue to accommodate larger substrates; this hypothesis could not be verified in CfusAmDH as the P5-mutant is unstable".

As a suggestion: one option could be to take one of the "Ala151" stable enzymes and try to mutate the position 151 with a glycine. If the mutation is tolerated, even bulkier substrates should be in principle accommodated.

We have already tried mutations into glycine at this position in different nat-AmDHs but, as mentioned but not detailed in our previous paper (Ducrot, L. *et al.* Expanding the substrate scope of native Amine dehydrogenases through *in silico* structural exploration and targeted protein engineering. *ChemCatChem* **14**, (2022)), it did not lead to notably better activities towards longer substrates compared to alanine mutants. Therefore we did not study these mutations further. Nevertheless, we will consider performing it in MetDB-02 and/or MetDB-03 in case they behave differently. This work will be part of the future publication dealing with

detailed characterization and biocatalysis application of some of these newly identified nat-AmDHs (see answer to comment 6).

10) Consider the sentence: "The ketones bearing the carbonyl function at the third carbon atom of the chain length (3C-ketones) was another class of target substrates. Even now, these are still poorly described in reductive amination even by other classes of enzymes such as RedAms, IREDs or eng-AmDHs, the only reported examples leading to the formation of the (R)-enantiomer with low enantiomeric excesses [cit. 38]"

This is not correct. This type of more bulky-bulky ketones for amination have been described for AmDHs (producing the R-enantiomer) in a few more other publications:

a) Knaus, T.; Böhmer, W.; Mutti, F. G., Amine dehydrogenases: efficient biocatalysts for the reductive amination of carbonyl compounds. *Green Chem.* 2017, 19, 453-463.

b) Wang, D.-H.; Chen, Q.; Yin, S.-N.; Ding, X.-W.; Zheng, Y.-C.; Zhang, Z.; Zhang, Y.-H.; Chen, F.-F.; Xu, J.-H.; Zheng, G.-W., Asymmetric Reductive Amination of Structurally Diverse Ketones with Ammonia Using a Spectrum-Extended Amine Dehydrogenase. *ACS Catal.* 2021, 11 (22), 14274-14283.

Furthermore, amination of cyclic bulky-bulky ketones with AmDHs have been described here:

c) Lv, T.; Feng, J.; Chen, X.; Luo, Y.; Wu, Q.; Zhu, D.; Ma, Y., Desymmetric Reductive Amination of 1,3-Cyclopentadiones to Single Stereoisomer of β -Amino Ketones with an All-Carbon Quaternary Stereocenter by Engineered Amine Dehydrogenases. *ACS Catal.* 2023, 13 (7), 5053-5061.

Also, of possible interest for the discussion in the manuscript, are substrates with an OH in position 1, but still having similar structural feature (OH-CH₂-CO-R):

d) Ming, H.; Yuan, B.; Qu, G.; Sun, Z., Engineering the activity of amine dehydrogenase in the asymmetric reductive amination of hydroxyl ketones. *Catal. Sci. Technol.* 2022, 12 (19), 5952-5960.

e) Chen, F.-F.; Cosgrove, S. C.; Birmingham, W. R.; Mangas-Sanchez, J.; Citoler, J.; Thompson, M. P.; Zheng, G.-W.; Xu, J.-H.; Turner, N. J., Enantioselective Synthesis of Chiral Vicinal Amino Alcohols Using Amine Dehydrogenases. *ACS Catal.* 2019, 9 (12), 11813-11818.

I think that the paragraph should be modified and put into the perspective of the literature above that should also be cited in this regard (at least suggestions from a to c).

In this study we focused on acyclic compounds, the cyclic ones such as cyclohexanone/cyclopentanone derivatives being well accepted by nat-AmDHs and other reductive aminases. Therefore, we did not cite the proposed reference c dealing with substituted 1,3-cyclopentadione. We have added the adapted references a) and b) to provide more exhaustive examples of studies describing the formation of (R)-amines from linear 3C-ketones. Despite their being less bulky substrates, hydroxymethyl ketones aminated by these enzymes are featured in the proposed references c) and d), in addition to our own work on

these substrates (Ducrot et al 2021 “Biocatalytic Reductive Amination by Native Amine Dehydrogenases to Access Short Chiral Alkyl Amines and Amino Alcohols”).

Finally, the authors correctly emphasize that they produce the opposite enantiomer (S) as the point of novelty. This was properly stated: “To the best of our knowledge, these enzymes are the only ones yet reported to provide the (S)-enantiomer of such 3C ketones (....)” This sentence could probably be moved within the previous paragraph to make clear to the reader this element of novelty earlier in the text.

To address this comment, the sentence has been modified in this paragraph to “Again, (S)-stereoselectivity predominates, but is not exclusively observed for some enzymes.”, and another sentence has been added to highlight this characteristic: “This (S)-stereoselectivity, already observed for members of this family, is a characteristic maintained in this extended group, differentiating these enzymes from the other NAD(P)-dependent enzymes performing reductive amination.”

11) Regarding the amination of ketones with amine donors larger than ammonia, I could not find the absolute configuration of the formed amine products (neither in main manuscript nor in Supplementary Figure 25). Were they (S)-configured?

In this paper, we described the activity toward other amine donors than ammonia only with cyclohexanone, so the corresponding amines are not chiral at the created C-sp³ carbon. In further studies, mentioned in points 6 and 9, we will check the (S)-stereochemistry of amines formed with prochiral ketones and alkylamines.

12) Regarding the amination of ketones with amine donors larger than ammonia using AmDHs, there is another publication prior the cited ref. 40 (NfRedAm or AdRedAm) that is:

Tseliou, V.; Masman, M. F.; Böhmer, W.; Knaus, T.; Mutti, F. G., Mechanistic Insight into the Catalytic Promiscuity of Amine Dehydrogenases: Asymmetric Synthesis of Secondary and Primary Amines. ChemBioChem 2019, 20 (6), 800-812.

This work could also be placed in the context of this discussion besides ref. 40. Here, the amines were enantioenriched (R). I assume that the authors got the (S) with their enzymes. This feature could be compared and highlighted.

We thank the reviewer for pointing out this reference, which is worth quoting for comparison. We have added the reference and modified the sentence as follows: “These enzymes could be complementary to NfRedAm, AdRedAm, Ch1-AmDH and Rs-AmDH, which mainly form (R)-methyl/ethylamines, even if their activities towards aromatic and acyclic aliphatic ketones, reported to be transformed by the latter, remain to be studied”

13) For the reasons explained previously, the sentence in the conclusion: “Moreover, activity towards C3-ketones, rarely reported for biocatalysts performing reductive amination, was found to be (....)” should be modified.

This sentence has been deleted in the more concise revised manuscript.

14) The next sentence in the conclusion should also be modified and placed better within the state-of-the-art: "(...) moderate to high activity for both alkylamine (methylamine and cyclopropylamine) and ammonia was verified for some enzymes, overcoming the restriction to primary amines and giving access to secondary amines".

The already described activity toward methylamine was very poor and only reported in our previous paper in a preliminary fashion (Mayol et al. 2019). The high conversions reported in this revised manuscript with methylamine and cyclopropylamine with various enzymes is a new result. The restriction to primary amines of previously studied members of the nat-AmDH family has been added to the state-of-the-art in the introduction, in the sentence "These enzymes are (*S*)-stereoselective [...] and active toward primary amines."

15) Supplementary Figure 5: please, indicate the band of the expressed enzyme (e.g., red circle) on the gel. The bands of the expressed enzymes are not well visible due to the moderate expression level. This could also be done for the other two gels in SI fig. 6 and 12 although here some bands are a lot more visible already (but others not).

The denatured protein sizes are in the narrow range of 38 - 41 kDa. Some arrows have been added at the relevant position on each of the gels illustrated in Supplementary Fig. 5, 6 and 8, on the left and the right. The bands of the overexpressed enzymes were not framed for each enzyme because this would obscure the other bands and further disrupt the visibility of the gel. For Figure 5, which displays gels for G1 enzymes, the overexpression was indeed not seen for many enzymes. This was already indicated in Supplementary Data 2 in columns 'gel induction' and 'gel crude cell lysate', in addition to Supplementary Data 3-4 for other groups. In those studies with a generic protocol, the amount of overexpressed enzyme in the soluble fraction can be low. Nevertheless, the sensitive analytic method can enable detection of low activity in the case of very low amounts of enzymes (and/or low specific activity), as exemplified by the enzyme in A7. However, we agree that some positive enzymes can be missed in this case.

Reviewer #3 (Remarks to the Author):

This manuscript describes an expansion of the native amine hydrogenase (nat-AmDH) family from 2,011 sequences (presumably in the UniProt database in 2020?) to 17,959 sequences from ten genome and metagenome databases (from 2019, 2020, 2021). The expanded family was then subjected to screening of representative members to provide a description of the substrate specificities; not surprisingly, the authors discovered novel specificities that would be useful in using members of the family to provide access to novel chiral amines.

The expanded membership was discovered using a new HMM for the C-terminal catalytic domain sequences of the original set of sequences. The sequences for the N-terminal Rossmann-fold domain were excluded to eliminate false positives, although the sequences identified using the new HMM for the catalytic domain were required to contain an N-terminal Rossmann-fold domain. This strategy is, in the opinion of this reviewer, the most generally instructive part of the manuscript.

Despite the length of the manuscript, many of the details of this expansion strategy are difficult to find. Most basic, the authors do not state where/how the original 2,011 sequences were identified? UniProt? Which Pfam or InterPro family?

The 2,011 original sequences were identified from UniProt (Mayol et al. 2019). At that time, AmDHs were automatically annotated as dihydrodipicolinate reductase, only based on their N-terminus domain (PF01113 / IPR000846). Their C-terminus domain was only annotated in April 2021 with the new Pfam entry (PF19328 / IPR045760), designed by A. Bateman (Pfam curator).

And, it is somewhat frustrating that the databases that were used are quite dated--2019, 2020, 2021. As a result, the number of sequences is now much larger?

Indeed, publicly available databases have been enriched since our selection, resulting in a much larger number of sequences. However, our strategy could be simply reused to search for new enzyme candidates in these updated databases. In this publication, we wanted to combine both *in silico* and *in vitro* studies to validate experimentally the activity of the family, which inevitably led to a time delay, enzyme production and activity tests being more time-consuming.

This manuscript does not mention AlphaFold structures. I suspect this is not surprising, given that the downstream enzymological characterization work presumably required significant time and effort. But, readers now would expect that a manuscript describing the expansion of a Pfam-curated family (?) would involve only sequence (HMM) methods. This strategy likely will not identify more divergent members of the family that would be expected to have additional novel substrate specificities.

We searched for a structural pattern ("catalophore") of the AmDH active site among the AlphaFold predictions, but without success. One limitation was the open-form structure, systematically obtained using the AlphaFold algorithm, for which the active site was wider than in the reference structures (on which the pattern was designed), and which did not yield any results. Otherwise, obtaining closed structures for the missing models would require significant

computing resources (e.g. molecular dynamics to force the closing state). In addition, the 9,886 models, obtained by MODELLER from the non-redundant set of 17,039 AmDH sequences, reasonably cover the corresponding phylogenetic tree, and in particular groups G1-G4 - G5, which are a group of enzymes that are assumed to be non-functional because they lack the critical P3 glutamate residue (as revealed by ASMC analysis).

This structural approach was already mentioned in the conclusion in the first submission but it is now described in more detail at the end of the section "Environmental sampling, clustering and analysis" (cf. comment 4.a) - Reviewer 1).

REVIEWERS' COMMENTS

Reviewer #1 (Remarks to the Author):

All comments and suggestions were adequately addressed - Green light by me.

Reviewer #2 (Remarks to the Author):

I confirm my evaluation regarding significance, novelty and impact of the work. In addition, the authors have significantly improved the readability of the manuscript by following the suggestions from all the reviewers. My comments from the previous round of revision were addressed either through changes in the main manuscript and SI, or in the rebuttal letter.

From my side, there is only one very minor comment for a correction before acceptance by Nature Communications.

In the introduction, change "nicotinamide cofactors" with "nicotinamide adenine dinucleotide cofactors".

Reviewer #3 (Remarks to the Author):

The revised manuscript addresses my comments and now can be accepted for publications.